# To Clip or not to Clip: the Dynamics of SGD with Gradient Clipping in High-Dimensions

**Noah Marshall, Ke Liang Xiao & Elliot Paquette**
McGill University
{noah.marshall2,keliang.xiao}@mail.mcgill.ca
elliot.paquette@mcgill.ca

**Atish Agarwala**
Google Deepmind
thetish@google.com

## Abstract

The success of modern machine learning is due in part to the adaptive optimization methods that have been developed to deal with the difficulties of training large models over complex datasets. One such method is gradient clipping: a practical procedure with limited theoretical underpinnings. In this work, we study clipping in a least squares problem under streaming SGD. We develop a theoretical analysis of the learning dynamics in the limit of large intrinsic dimension—a model and dataset dependent notion of dimensionality. In this limit we find a deterministic equation that describes the evolution of the loss and demonstrate that this equation predicts the path of clipped SGD on synthetic, CIFAR10, and Wikitext2 data. We show that with Gaussian noise clipping cannot improve SGD performance. Yet, in other noisy settings, clipping can provide benefits with tuning of the clipping threshold. We propose a simple heuristic for near optimal scheduling of the clipping threshold which requires the tuning of only one hyperparameter. We conclude with a discussion about the links between high-dimensional clipping and neural network training.

## 1 Introduction

Stochastic gradient descent (SGD) methods are the standard for nearly all large scale modern optimization tasks. Even with the ever growing complexities of neural nets, with sufficient hyperparameter tuning, SGD often outperforms other more complex methods. To deal with the difficulties of training large models over complex datasets, adaptive SGD methods have been developed. One of the simplest such methods is gradient clipping (Mikolov, 2012; Pascanu et al., 2013). Gradient clipping replaces any stochastic gradient $\nabla_{\boldsymbol{\theta}} f_{\boldsymbol{\theta}}(\mathbf{x})$ with a clipped gradient $\mathrm{clip}_c(\nabla_{\boldsymbol{\theta}} f_{\boldsymbol{\theta}}(\mathbf{x}))$, for some threshold $c$, where

$$\mathrm{clip}_c(\mathbf{z}) = \min\left(1, \frac{c}{\|\mathbf{z}\|}\right)\mathbf{z}. \tag{1}$$

While gradient clipping was first introduced to address the problem of exploding gradients in recurrent neural networks, it has become an integral part of training models for NLP gpt3. It has also found use in other domains such as differential privacy (Abadi et al., 2016; Pichapati et al., 2019) and computer vision (Tolstikhin et al., 2021; Dosovitskiy et al., 2021).

Despite widespread use, the reasons behind the effectiveness of clipping remain somewhat a mystery. For instance, it is unclear exactly how the gradient distribution affects training, or for which distributions clipping can offer benefits. It is hypothesized that the distribution of the gradient norms plays a large role (Zhang et al., 2020b). Also, it is unknown how one should adjust the clipping threshold as the problem scales. There has been growing interest in how models and their optimal hyper-parameters scale with dimension (Kaplan et al., 2020). Understanding this behaviour would allow one to perform hyperparameter tuning on smaller, more efficient models before scaling to a potentially very large final architecture.

In this work we develop a theory of clipped SGD in high-dimensions under the mean-squared error loss (MSE) over a class of random least-squares problems. After formally introducing the class of considered problems (Sec. 2), we show the following:

- In high-dimensions the dynamics of clipped SGD (C-SGD) are well described by an SDE, clipped homogenized SGD (C-HSGD). We provide a *non-asymptotic* bound on the difference of the risk curves between C-SGD and C-HSGD. Under C-HSGD the risk evolution can be described by a system of ODEs (Sec. 3) which we demonstrate predict the training dynamics under real-world data.

- Using C-HSGD, we show that the differences between clipped and unclipped SGD can be described by two unitless *reduction factors* $\mu$ and $\nu$ which encode the effect of clipping (Sec. 3).

- The reduction factors control the stability of the algorithm. They describe the precise clipping-learning rate combinations which are convergent. Moreover, we identify some clipping schedules that improve stability (Sec. 4).

- We find a general criterion for when clipping can speed up optimization, described by a different ratio of the reduction factors. We then identify a problem setup where clipping never helps as well as one where clipping improves performance. (Sec. 5).

- Using these insights, we propose a simple, heuristic clipping and learning rate schedule. This schedule is shown to closely follow an optimal schedule which we prove will *never* underperform SGD.

We conclude with a discussion about the links between our analysis and quantities measurable in real neural networks.

**Related work:** The distribution of noise in stochastic gradients and its effect on training was studied by (Zhang et al., 2020b). They argue that this noise is well approximated by a Gaussian for ResNets (He et al., 2015) trained on Imagenet (Deng et al., 2009), while a heavy-tailed distribution is more appropriate with BERT (Devlin et al., 2019) on an NLP dataset. They show that for heavy-tailed noise unclipped SGD diverges while clipped SGD can converge. Other theoretical analyses on clipping often focus on imposing smoothness conditions on the loss function, and then performing analysis for fixed learning rates (Koloskova et al., 2023; Zhang et al., 2020a; Chen et al., 2020). These works have shown that fixed rate clipped SGD can outperform unclipped SGD under certain conditions. Other works have also studied SGD through the lens of SDEs (Li et al., 2017; Mandt et al., 2016; Barrett & Dherin, 2021). More recently, partly spurred by the sheer size of modern models as well as the apparent regularity at which they scale (Kaplan et al., 2020), there has been an interest in studying stochastic optimization in high-dimensions with SDEs. There is a formal correspondence between the dynamics of learning-relevant quantities like the loss and the trajectory of an equivalent SDE. These relationships have been worked out for SGD in the streaming setup over a variety of losses (Collins-Woodfin et al., 2023; Paquette et al., 2022), and the resulting analyses lead to quantities which can be useful for understanding learning dynamics in practical models (Agarwala et al., 2023).

## 2 PROBLEM SETUP

In this work, we consider linear regression using the mean-squared loss

$$\mathcal{L}(\boldsymbol{\theta}, \mathbf{x}, y) = \| \langle \mathbf{x}, \boldsymbol{\theta} \rangle - y \|^2 / 2, \tag{2}$$

in the streaming or one-pass scenario, where data is not reused. Clipped SGD (C-SGD), without mini-batching, is described by the iteration

$$\boldsymbol{\theta}_{k+1} = \boldsymbol{\theta}_k - \eta_k \operatorname{clip}_{c_k} \left( \nabla_{\boldsymbol{\theta}} \mathcal{L}(\boldsymbol{\theta}_k, \mathbf{x}_{k+1}, y_{k+1}) \right), \tag{3}$$

where $\nabla_{\boldsymbol{\theta}} \mathcal{L}(\boldsymbol{\theta}_k, \mathbf{x}_{k+1}, y_{k+1}) = (\langle \mathbf{x}_{k+1}, \boldsymbol{\theta}_k \rangle - y_{k+1}) \mathbf{x}_{k+1}$ with initialization $\boldsymbol{\theta}_0 \in \mathbb{R}^d$. We assume that the samples $\{(\mathbf{x}_k, y_k)\}_{k \geq 0}$, consisting of data $\mathbf{x}_k$ and targets $y_k$, satisfy the following:

**Assumption 1.** *The data $\mathbf{x} \in \mathbb{R}^d$ are Gaussian with covariance $\mathbf{K}$. The targets $y$ are generated by $y = \langle \mathbf{x}, \boldsymbol{\theta}^* \rangle + \epsilon$, where $\epsilon$ represents noise and $\boldsymbol{\theta}^*$ is the ground-truth.*

*The noise is centered and subgaussian (see Vershynin (2018) for more details) with subgaussian norm $\|\epsilon\|_{\psi_2} \leq v$ and variance $\mathbb{E}[\epsilon^2] = \sigma^2$ for some $v, \sigma \geq 0$.*

We formulate a more general version of our results for non-Gaussian data in Appendix A.

**Definition 1.** *Define the population risk and the noiseless risk:*

$$\mathcal{P}(\boldsymbol{\theta}) = \mathbb{E}_{(\mathbf{x}, \epsilon)} \left[ (\langle \mathbf{x}, \boldsymbol{\theta} - \boldsymbol{\theta}^* \rangle - \epsilon)^2 \right]/2 \quad \text{and} \quad \mathcal{R}(\boldsymbol{\theta}) = \mathbb{E}_{\mathbf{x}} \left[ \langle \mathbf{x}, \boldsymbol{\theta} - \boldsymbol{\theta}^* \rangle^2 \right]/2, \tag{4}$$

*as well as the distance to optimality*

$$\mathcal{D}(\boldsymbol{\theta}) = \|\boldsymbol{\theta} - \boldsymbol{\theta}^*\|^2.$$

Our theory is phrased in terms of the intrinsic dimension, a statistical notion of dimensionality which is occasionally much smaller than the ambient dimension $\mathfrak{d}$. There are interesting settings where these dimensions are effectively interchangeable, and as such, the reader may wish to, at first glance, consider the results to be phrased in terms of the ambient dimension.

**Definition 2** (Intrinsic Dimension). *Let the data* $\mathbf{x} \in \mathbb{R}^{\mathfrak{d}}$ *have covariance matrix* $\mathbf{K}$. *Define the intrinsic dimension of the data to be*

$$d = \mathrm{Tr}(\mathbf{K})/\|\mathbf{K}\|, \tag{5}$$

*where* $\|\mathbf{K}\|$ *refers to the operator norm. Note that* $d \leq \mathfrak{d}$. *We will refer to* $\mathfrak{d}$ *as the ambient dimension.*

**Assumption 2.** *The covariance matrix* $\mathbf{K}$ *is normalized such that* $\|\mathbf{K}\| = 1$. *Note that this assumption may always be satisfied by rescaling the problem.*

The definition of $d$ can be extended to and measured in real neural networks trained on real datasets; see Appendix C.3 for more details.

We allow for the scheduling of both the clipping threshold and the learning rate. Specifically,

**Assumption 3.** *There are continuous bounded functions* $\eta : \mathbb{R}^+ \to \mathbb{R}^+$ *and* $c : \mathbb{R}^+ \to \mathbb{R}^+$ *such that*

$$c_k = c(k/d)\sqrt{d} \qquad\qquad \eta_k = \eta(k/d)/d. \tag{6}$$

We note that while it is reasonable for $c(t) = \infty$ (which is to say that no clipping occurs), for technical reasons, we shall not allow this in our main theorem.

## 3 CLIPPED HOMOGENIZED SGD

Our main result shows that the risk of C-SGD is well-approximated by the solution to an SDE which we call *clipped homogenized SGD* (C-HSGD):

**Definition 3** (Clipped Homogenized SGD). *Denote the stochastic gradient as* $\ell_{\boldsymbol{\theta}} \mathbf{x}$, *where* $\ell_{\boldsymbol{\theta}} = \langle \mathbf{x}, \boldsymbol{\theta} - \boldsymbol{\theta}^* \rangle - \epsilon$. *Define the descent reduction factor and the variance reduction factor*

$$\mu_c(\boldsymbol{\theta}) = \frac{\|\mathbb{E}[\mathrm{clip}_c(\ell_{\boldsymbol{\theta}})\mathbf{x}]\|}{\|\mathbb{E}[\ell_{\boldsymbol{\theta}}\mathbf{x}]\|} \qquad \text{and} \qquad \nu_c(\boldsymbol{\theta}) = \frac{\mathbb{E}[\mathrm{clip}_c^2(\ell_{\boldsymbol{\theta}})]}{\mathbb{E}[\ell_{\boldsymbol{\theta}}^2]}. \tag{7}$$

*Then C-HSGD is defined to be the solution to*

$$\mathrm{d}\boldsymbol{\Theta}_t = -\eta(t)\mu_{c(t)}(\boldsymbol{\Theta}_t)\nabla\mathcal{P}(\boldsymbol{\Theta}_t)\mathrm{d}t + \eta(t)\sqrt{\frac{2\nu_{c(t)}(\boldsymbol{\Theta}_t)\mathcal{P}(\boldsymbol{\Theta}_t)\mathbf{K}}{d}}\mathrm{d}\mathbf{B}_t, \tag{8}$$

*where initialization is taken to be the same as SGD and* $B_t$ *is a standard Brownian motion.*

**Remark:** If the noise is heavy-tailed ($\sigma = \infty$), we can reformulate the above by redefining $\nu_c(\boldsymbol{\theta}) = \mathbb{E}[\mathrm{clip}_c^2(\ell_{\boldsymbol{\theta}})]$ and changing the coefficient on the Brownian motion term to $\eta(t)\sqrt{\nu_{c(t)}(\boldsymbol{\Theta}_t)\mathbf{K}/d}$. This comes from noticing that $\mathbb{E}[\ell_{\boldsymbol{\theta}}^2] = 2\mathcal{P}(\boldsymbol{\theta})$ and $\mathbb{E}[\mathrm{clip}_c^2(\ell_{\boldsymbol{\theta}})] < \infty$ even in the heavy-tailed case due to the effects of clipping. This yields the immediate observation that clipping improves SGD under heavy-tailed noise since clipped SGD converges while SGD does not.

C-HSGD has similar structure to an SDE previously established for unclipped SGD (Paquette et al., 2022), with the addition of reduction factors $\mu_c$ and $\nu_c$ that capture the effects of clipping. In essence, the homogenized SGD suggests that, clipped SGD is still driven by a drift term in the direction of $-\nabla_{\boldsymbol{\theta}}\mathcal{R}$—but clipping provides a bias *against* the gradient which shrinks the descent term (hereafter

the negative term in (8)). Meanwhile, the diffusion term is shrunk by $\nu_c(\theta)$ due to the reduction of variance of the clipped gradients. These terms imply a tradeoff: clipping should aim to reduce variance (decrease $\nu_c$) more than it shrinks the descent term (decrease $\mu_c$). We investigate this trade-off in detail in Section 5.

We compute $\mu_c$ and $\nu_c$ for select data and noise distributions (Appendix D). For Gaussian data, Stein's Lemma gives us a form for $\mu_c$ which can be interpreted as the probability of *not* clipping:

$$\mu_c(\boldsymbol{\theta}) = \mathbb{P}(|\ell_{\boldsymbol{\theta}}| \leq c). \tag{9}$$

For non-Gaussian data the exact form of $\mu_c$ is challenging to compute due to its dependence on $\mathbf{x}$; however, Equation (9) can give a good approximation in many high-dimensional settings due to Central Limit Theorem effects. As an example, risk curves for linear models trained on CIFAR10 (Figure 1 (c)) and Wikitext2 (Figure 1 (d)) are well-captured by the dynamics of C-HSGD with $\mu_c$ computed using the Gaussian form from Equation 9. [1] See the experiment details in Appendix H.

This form can also be used to tractably compute $\mu_c$ in the non-linear setting using information already computed during C-SGD. See Appendix C.2 for a full description of the extension of $\mu_c$ and $\nu_c$ to this setting, as well as a demonstration of the technique during training of ResNet18 and ViT on CIFAR10. This allows us to cheaply estimate $\mu_c$ and $\nu_c$ for many values of $c$ through training.

There exist a set of tractable lower and upper bounds for $\mu_c$ and $\nu_c$ which can be expressed in terms of the risk. In Appendix E, we show that for all noise distributions with well-behaved densities that are positive near 0, there exist positive constants $\kappa_l$ and $\kappa_u$ such that

$$\kappa_l \min\left(1, \frac{c}{\sqrt{2\mathcal{R}(\boldsymbol{\theta}) + \sigma^2}}\right) \leq \mu_c(\boldsymbol{\theta}) \leq \kappa_u \min\left(1, \frac{c}{\sqrt{2\mathcal{R}(\boldsymbol{\theta}) + \sigma^2}}\right) \tag{10a}$$

$$\kappa_l \min\left(1, \frac{c^2}{2\mathcal{R}(\boldsymbol{\theta}) + \sigma^2}\right) \leq \nu_c(\boldsymbol{\theta}) \leq \kappa_u \min\left(1, \frac{c^2}{2\mathcal{R}(\boldsymbol{\theta}) + \sigma^2}\right). \tag{10b}$$

This simple form can be used to derive simple clipping schedules for MSE loss, which we explore in Section 5.3.

We now state our main theorem, which formalizes how C-HSGD is a good description of C-SGD:

**Theorem 1.** *Suppose that Assumptions 1, 2 and 3 hold. Suppose that $\boldsymbol{\Theta}_t$ and $\boldsymbol{\theta}_k$ are independent realizations of C-HSGD and C-SGD with equal, deterministic initial conditions. Let $\bar{c} = \sup_t c(t)$ and $\bar{\eta} = \sup_t \eta(t)$. There is a constant $\mathcal{C} = \mathcal{C}(\mathbf{v}, (n/d), \bar{c}, \bar{\eta}, \|\boldsymbol{\theta}_0 - \boldsymbol{\theta}^*\|^2)$, a stochastic process $\mathcal{E}$, and a constant $m = m(\mathbf{v})$ so that for any $1 \leq u \leq md$ and any $n$*

$$\sup_{0 \leq k \leq n} \left\| \begin{bmatrix} \mathcal{R}(\boldsymbol{\theta}_k) \\ \mathcal{D}(\boldsymbol{\theta}_k) \end{bmatrix} - \begin{bmatrix} \mathcal{R}(\boldsymbol{\Theta}_{k/d}) \\ \mathcal{D}(\boldsymbol{\Theta}_{k/d}) \end{bmatrix} \right\| \leq \mathcal{C}\mathcal{E}(n/d)u\log(d)d^{-1/2}, \tag{11}$$

*with probability $1 - e^{-u}$ and provided the right hand side is less than $1$. The stochastic process $\mathcal{E}$ is given by*

$$\mathcal{E}(t) = \exp\left( \int_0^t \frac{C\eta(s)^2\sigma \, \mathrm{d}s}{\sqrt{\mathcal{R}(\boldsymbol{\Theta}_s) + \mathcal{R}(\boldsymbol{\theta}_{\lfloor sd \rfloor})}} \right)$$

*for an absolute constant $C > 0$. The constant $\mathcal{C}$ can be bounded by*

$$\mathcal{C} \leq C\sqrt{n/d}\,\bar{\eta}\mathbf{v}^2 \cdot ((1 + \|\boldsymbol{\theta}_0 - \boldsymbol{\theta}^*\|^2)\mathbf{v}^2 + \bar{c}^2\sqrt{n/d}) \cdot \exp\left(C\max\{\bar{\eta}, \bar{\eta}^2\}(n/d)\right).$$

Informally, this theorem says that

$$\sup_{0 \leq k \leq n} \left\| \begin{bmatrix} \mathcal{R}(\boldsymbol{\theta}_k) \\ \mathcal{D}(\boldsymbol{\theta}_k) \end{bmatrix} - \begin{bmatrix} \mathcal{R}(\boldsymbol{\Theta}_{k/d}) \\ \mathcal{D}(\boldsymbol{\Theta}_{k/d}) \end{bmatrix} \right\| = \mathcal{O}(\log(d)d^{-1/2}).$$

In particular, as $d$ grows, the risk curves of C-SGD and C-HSGD look closer to one another for longer time windows and with higher probability. Under additional assumptions,[2] the C-HSGD curve converges to a dimension-independent deterministic limit and thus so does clipped SGD. We discuss how to find this deterministic limit in the following section.

The complete proof is detailed in Appendix B along with the theorem statement and proof for non-Gaussian data.

---

[1]Code to reproduce the results is available at https://github.com/nmarzz/clip.

[2]The spectrum of $\mathbf{K}$ converges and the initialization $\boldsymbol{\theta}_0 - \boldsymbol{\theta}^*$ converges

**Extracting deterministic dynamics:** For any twice differentiable function $q$, the C-HSGD dynamics can be expressed as follows:

$$\mathrm{d}q(\boldsymbol{\Theta}_t) = -\eta(t)\mu_{c(t)}(\boldsymbol{\Theta}_t)\nabla\mathcal{P}(\boldsymbol{\Theta}_t)^T\nabla q(\boldsymbol{\Theta}_t)\mathrm{d}t + \frac{\eta(t)^2}{d}\nu_{c(t)}(\boldsymbol{\Theta}_t)\mathcal{P}(\boldsymbol{\Theta}_t)\operatorname{Tr}(\mathbf{K}\nabla^2 q)\mathrm{d}t + \mathrm{d}\mathcal{M}_t, \tag{12}$$

where $\mathcal{M}_t$ is a martingale term that vanishes as $d \to \infty$. This is an example of the concentration of measure phenomenon observed in high-dimensional probability. As a result, we obtain a good deterministic approximation for the evolution of $q(\boldsymbol{\Theta}_t)$ by setting $\mathcal{M}_t \equiv 0$.

Using this idea, we introduce deterministic equivalents, $R_t$ and $D_t$, for the risk terms $\mathcal{R}(\boldsymbol{\Theta}_t)$ and $\mathcal{D}(\boldsymbol{\Theta}_t)$, respectively. Solving for $R_t$ and $D_t$ involves forming and solving a coupled system of $d$ ordinary differential equations. Additionally, Theorem 1 implies that these risks under clipped stochastic gradient descent converge to their deterministic equivalents. The complete details and derivation of this system of ODEs along with the statement and proof of C-SGD's convergence to the deterministic equivalents are available in Appendix G.

A numerical comparison of C-SGD, C-HSGD, and the ODEs is provided in Figure 1. In this figure, we demonstrate that these dynamics describe the risk of clipped SGD with both Cauchy and Gaussian distributed noise, despite the fact that unclipped SGD does not converge under Cauchy noise. In the same figure we show that these deterministic equivalents describe the dynamics of real-world data.

A simple example of the above theory can be had when the covariance of the data is the identity matrix.

**Example 1** (Isotropic data). *When the data is isotropic Gaussian, the risk $R_t$ solves the autonomous ODE:*

$$\dot{R}_t = -2\eta(t)\mu_{c(t)}R_t + \eta(t)^2\nu_{c(t)}(R_t + \sigma^2/2), \tag{13}$$

*where $R_0 = \mathcal{R}(\boldsymbol{\Theta}_0)$.*

Since the data is Gaussian, it is possible to express $\mu_c$ and $\nu_c$ as functions of the risk. As a slight abuse of notation we shall also write $\mu_c(\mathcal{R}(\boldsymbol{\theta}))$ or $\nu_c(\mathcal{R}(\boldsymbol{\theta}))$ for $\mu_c(\boldsymbol{\theta})$ or $\nu_c(\boldsymbol{\theta})$ respectively. We will suppress the dependence on $\boldsymbol{\theta}$ where appropriate, as we have in Example 1.

## 4 STABILITY ANALYSIS

In this section we establish stability conditions for streaming SGD with clipping. Stability thresholds from convex models are useful for understanding dynamics in deep learning (Cohen et al., 2021; Agarwala et al., 2023). Additionally, a larger range of stable learning rates can prevent failures in costly training runs. We show that the largest stable learning rate is structurally similar to that of the unclipped SGD case, but with the introduction of the reduction factors $\mu_c$ and $\nu_c$ which account for the effects of clipping.

From Equation (12), we observe that for either the risk $\mathcal{R}$ or the distance to optimality $\mathcal{D}$, the instantaneous time derivative is quadratic in the learning rate $\eta(t)$. This implies that we can compute a stability threshold for the learning rate, determining whether, in high-dimensions, these measures of suboptimality increase or decrease. We find the critical values $\eta_{\mathcal{R}}^*(t)$ and $\eta_{\mathcal{D}}^*(t)$ such that $\mathbb{E}[\mathrm{d}\mathcal{R}(\Theta_t)] = 0$ and $\mathbb{E}[\mathrm{d}\mathcal{D}(\Theta_t)] = 0$. In particular,

$$\eta_{\mathcal{R}}^*(t) = \frac{d\|\nabla\mathcal{P}(\boldsymbol{\Theta}_t)\|^2}{\operatorname{Tr}(\mathbf{K}^2)\mathcal{P}(\boldsymbol{\Theta}_t)}\frac{\mu_{c(t)}(\boldsymbol{\Theta}_t)}{\nu_{c(t)}(\boldsymbol{\Theta}_t)} \quad \text{and} \quad \eta_{\mathcal{D}}^*(t) = \frac{\mathcal{R}(\boldsymbol{\Theta}_t)}{\mathcal{P}(\boldsymbol{\Theta}_t)}\frac{\mu_{c(t)}(\boldsymbol{\Theta}_t)}{\nu_{c(t)}(\boldsymbol{\Theta}_t)}. \tag{14}$$

This implies that clipping increases instantaneous stability (for both $\mathcal{R}$ and $\mathcal{D}$) relative to unclipped SGD when

$$\frac{\mu_{c(t)}(\boldsymbol{\Theta}_t)}{\nu_{c(t)}(\boldsymbol{\Theta}_t)} > 1. \tag{CSC}$$

We refer to this as the clipped-stability-criterion (CSC). This can be interpreted as as a relative signal-to-noise-ratio; the fraction of clipped gradients $\mu$ reduces the signal, while clipping reduces the noise through the reduction factor $\nu$. Stability is increased when the relative signal-to-noise-ratio is greater than 1. Clipping significantly enhances stability when a small fraction of samples contribute disproportionately to the gradient norm.

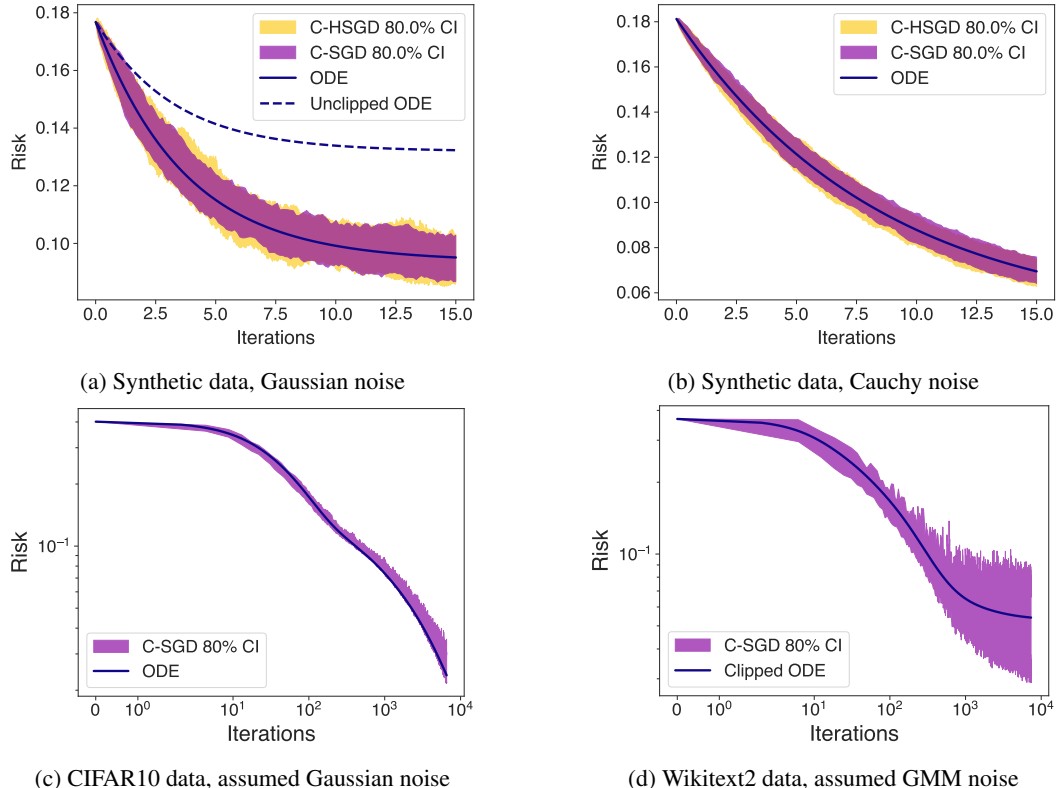

(a) Synthetic data, Gaussian noise          (b) Synthetic data, Cauchy noise

(c) CIFAR10 data, assumed Gaussian noise     (d) Wikitext2 data, assumed GMM noise

Figure 1: Comparison of C-SGD, C-HSGD and their deterministic equivalent (ODE) with Gaussian or Cauchy noise, from the Student-t family. The solution to the unclipped ODE for reference in the Gaussian case. Unclipped SGD does not converge under Cauchy noise. We also compare clipped SGD with CIFAR10 as well as Wikitext2 data. In all cases, the deterministic equivalent ODE closely match the actual path of clipped SGD. Experiment details are available in Appendix H.

Clipping will increase the stability of SGD with a small enough choice of $c$:

**Theorem 2.** *For data* $\mathbf{x} \sim N(0, \mathbf{K})$ *one may always choose the clipping schedule* $c$ *small enough to satisfy the* (CSC).

The proof of this theorem follows from an application of L'Hôpital's rule and is available in Appendix F. We provide plots of the (CSC) in Figure 2 under various settings. We conjecture that this result extends beyond Gaussian data, but the current intractability of $\mu$ for general data makes precise claims difficult.

Counterintuitively, clipping can also *decrease* stability in some cases, when the bias towards $\nabla_{\boldsymbol{\theta}} \mathcal{R}$ ($\mu$) is reduced more than the overall gradient norms ($\nu$). This shows that some care must be taken to avoid clipping being detrimental. The proof of the following theorem straightforwardly uses the definitions of $\mu_c$ and $\nu_c$ and can be found in Appendix F.

**Theorem 3.** *Consider* $\mathbf{x} \sim N(0, \mathbf{K})$ *and noise with the distribution given by*

$$\mathbb{P}(\epsilon = -\lambda) = p/2, \qquad \mathbb{P}(\epsilon = 0) = 1 - p, \qquad \mathbb{P}(\epsilon = \lambda) = p/2, \qquad (15)$$

*for* $p \in (0, 1)$ *and* $\lambda > 0$. *Then, there is a constant* $r$ *depending on* $p, \lambda$ *so that when* $R_t \leq r$ *there always exists* $c(t)$ *such that the* (CSC) *is less than* 1. *Therefore, clipped SGD can be less stable than unclipped SGD.*

## 5 WHEN DOES CLIPPED SGD OUTPERFORM UNCLIPPED SGD?

We now ask: under what settings can clipping improve the performance of SGD? Specifically, with the optimal learning rate schedule for unclipped SGD, does there exist a clipping-learning rate combination such that clipping achieves a lower loss at time $T$?

We will use Equation (12) to answer this question. We first present detailed calculations in the isotropic case to find an exact condition on the gradient distribution where clipping improves training. We then show that this condition still applies under anisotropic data. We provide examples and plots of this condition to develop intuition on when clipping helps to improve training. Finally, we propose a new, heuristic clipping schedule which very closely matches an optimal clipping schedule which we prove *never* underperforms unclipped SGD.

### 5.1 ISOTROPIC DATA

Consider the case of isotropic data where $\mathbf{x} \sim N(0, \mathbf{I})$. Define $R_t^\infty$ to be the deterministic equivalent of $\mathcal{R}(\varphi_t)$, where $\varphi_t$ is C-HSGD with $c(t) \equiv \infty$ (which is to say unclipped HSGD). Example 1 shows that $R_t$ and $R_t^\infty$ solve the following ODEs,

$$\frac{\mathrm{d}R_t}{\mathrm{d}t} = -2\eta(t)\mu_{c(t)}R_t + \frac{\eta^2(t)}{2}\nu_{c(t)}(2R_t + \sigma^2), \quad \frac{\mathrm{d}R_t^\infty}{\mathrm{d}t} = -2\eta(t)R_t^\infty + \frac{\eta^2(t)}{2}(2R_t^\infty + \sigma^2). \tag{16}$$

These results enable a comparison between clipped and unclipped SGD. Since these ODEs are quadratic in $\eta(t)$, it is straightforward to greedily maximize their instantaneous rate of descent, resulting in the globally optimal learning rate schedule. Optimizing each ODE over $\eta(t)$ yields

$$\frac{\mathrm{d}R_t}{\mathrm{d}t} = -\frac{R_t^2}{R_t + \sigma^2/2}\frac{\mu_{c(t)}^2}{\nu_{c(t)}} \qquad \frac{\mathrm{d}R_t^\infty}{\mathrm{d}t} = -\frac{(R_t^\infty)^2}{R_t^\infty + \sigma^2/2}. \tag{17}$$

When $R_t = R_t^\infty$, we see that the rate of descent is faster and thus clipping improves SGD exactly when there exists a $c(t)$ such that

$$\frac{\mu_{c(t)}^2(R_t)}{\nu_{c(t)}(R_t)} > 1. \tag{CCC}$$

We call this inequality the clipping-comparison-criterion (CCC). Therefore, in our setting we can exactly understand when clipping is helpful to training. Informally, the improvement criterion tells us that clipping is effective when it can *reduce* the variance of the gradient norms, via $\nu_{c(t)}$ more than it reduces the squared reduction to the descent term $\mu_{c(t)}^2$. This is consistent with previous observations that, in practice, clipping is effective when the distribution of the gradient norms is heavy-tailed Zhang et al. (2020b), but gives a quantitative rule for comparison. To give some intuition, we provide some plots of these thresholds over various types of noise distributions in Figure 2. In Appendix C.1 we provide similar plots involving the Student-t family of distributions. This family has a degrees of freedom parameter that allows the family to vary between Cauchy (df = 1) and Gaussian (as df goes to infinity). These plots provide an illustration of the effect that heavy-tailedness can have on the (CSC) and the (CCC).

### 5.2 ANISOTROPIC DATA

The previous results show that with isotropic data, the optimal clipping schedule can be found by maximizing the (CCC) at each time point. Inspired by this observation, we describe a procedure which, given a learning rate schedule for unclipped SGD, gives us a learning rate-clipping schedule pair which performs at least as well as unclipped SGD. We then provide a simple heuristic of this procedure, which experimentally achieves the same performance.

Consider a learning rate schedule $\eta(t)$, used to train unclipped SGD. We define the max-(CCC) clipping threshold schedule as follows: At step $t$, we first set the clipping threshold to

$$c^*(t) = \mathrm{argmax}_c \frac{\mu_c^2(R_t)}{\nu_c(R_t)}. \tag{18}$$

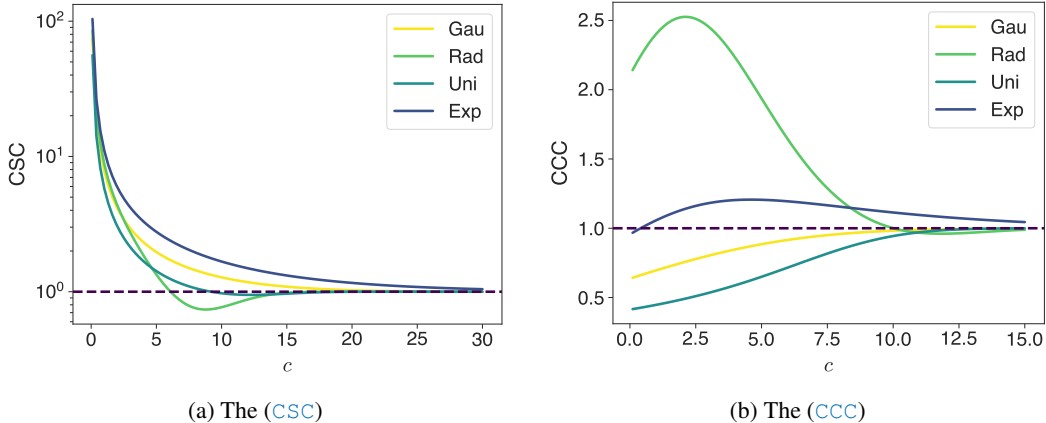

(a) The (CSC)            (b) The (CCC)

Figure 2: The (CSC) and (CCC) across various noise distributions: Gaussian (Gau), Rademacher-like (Rad), uniform on $[-M, M]$ (Uni), and symmetrized exponential (Exp) noise. The (CSC) is computed with $R = 3$, $\sigma = 9$, $p = 0.7$; the (CCC) figure uses $R = 3$, $\sigma = 5$, $p = 0.2$ (where $p$ is a parameter for Rademacher-like noise). Parameters are chosen to illustrate different behaviours.

Given a clipping threshold $c$, we define a *compensated* learning rate for clipped SGD by

$$\tilde{\eta}(t, c) = \eta(t)/\mu_c(R_t). \tag{19}$$

Effectively, this learning rate compensates for the fact that clipping biases SGD against the gradient such that clipped SGD now has the same instantaneous descent term as unclipped SGD. We now choose $\eta^*(t) = \tilde{\eta}(t, c^*(t))$ as our learning rate. In Appendix C.1 we provide plots of Equation (18) for various distributions.

This schedule will never underperform unclipped SGD. If the (CCC) is never satisfied we have $c^*(t) \equiv \infty$ and $\eta^*(t) = \eta(t)$, recovering the original, unclipped SGD. However, if the (CCC) is satisfied at any time the max-(CCC) schedule will take advantage of this and provide improvements to optimization. In order to show this, we first have to solve for $R_t$ under anisotropic data. In this setting, the risk is the sum of two parts: a gradient flow term and an integrated correction term. The gradient flow term is associated with the infinitesimal learning rate limit of SGD. It *decreases* the risk and comes from solving the underlying problem. The correction term arises because the actual learning rate is not infinitesimal. It encodes the errors made by SGD and *increases* the risk. Gradient flow is defined to be,

$$d\boldsymbol{\Phi}_t^{\text{gf}} = -\nabla \mathcal{R}(\boldsymbol{\Phi}_t^{\text{gf}}) \, dt, \tag{20}$$

with $\boldsymbol{\Phi}_0^{\text{gf}} = \boldsymbol{\theta}_0$. Then the gradient flow term is $\mathcal{R}(\boldsymbol{\Phi}_t^{\text{gf}})$. In Appendix G, we show $R_t$ with any learning rate $\eta(t)$ and clipping schedule $c(t)$ solves

$$R_t = \mathcal{R}(\boldsymbol{\Phi}_{\Gamma_T^c}^{\text{gf}}) + \frac{1}{d} \int_0^t \eta^2(s) \nu_{c(s)} \text{Tr}(\mathbf{K}^2 e^{-2\mathbf{K}(\Gamma_t^c - \Gamma_s^c)})(R_s + \sigma^2/2) ds, \tag{21}$$

where $\Gamma_t^c = \int_0^t \eta(s) \mu_{c(s)} ds$ is the clipped integrated learning rate. The integral term in Equation (21) is the finite learning rate correction. The risk of unclipped SGD can be computed using $c(t) \equiv \infty$:

$$R_t^\infty = \mathcal{R}(\boldsymbol{\Phi}_{\Gamma_T}^{\text{gf}}) + \frac{1}{d} \int_0^t \eta^2(s) \text{Tr}(\mathbf{K}^2 e^{-2\mathbf{K}(\Gamma_t - \Gamma_s)})(R_s^\infty + \sigma^2/2) ds. \tag{22}$$

where $\Gamma_t = \int_0^t \eta(s) ds$. This gives us the following theorem:

**Theorem 4.** *Given SGD with learning rate schedule $\eta(t)$ and clipped SGD with learning and clipping schedules $\eta^*(t)$ and $c^*(t)$, then $R_T \le R_T^\infty$. If there exists a $t \in [0, T]$ such that the (CCC) holds then $R_T < R_T^\infty$. Conversely, if $\mu_c^2(R)/\nu_c(R) \le 1$ for all $R > 0$ and $c > 0$, then for any learning and clipping schedules $\eta(t)$ and $c(t)$, SGD with the compensated learning rate schedule $\eta(t)\mu_{c(t)}$ has $R_T^\infty \le R_T$.*

*Proof.* With these choices, the clipped risk (21) solves

$$R_T = \mathcal{R}(\mathbf{\Phi}_{\Gamma_T}^{\mathrm{gf}}) + \frac{1}{d}\int_0^T \eta^2(s)\frac{\nu_{c(s)}}{\mu_{c(s)}^2}\operatorname{Tr}(\mathbf{K}^2 e^{-2\mathbf{K}(\Gamma_T - \Gamma_s)})(R_s + \sigma^2/2)\mathrm{d}s. \tag{23}$$

Note that the gradient flow term is identical to that of unclipped SGD. Since the (CCC) is satisfied, the integrated correction term is no larger than unclipped SGD and thus $R_T \le R_T^\infty$. If the (CCC) occurs at some $t$, then in fact $R_T < R_T^\infty$. For the converse, one substitutes the learning rate schedule into (22) and sees it is smaller than (21). □

We note that this result holds for any choice of the unclipped learning rate, even the optimal one. Therefore, if the (CCC) holds at some point along the optimal unclipped SGD trajectory then the benefits of gradient clipping cannot be matched by unclipped SGD.

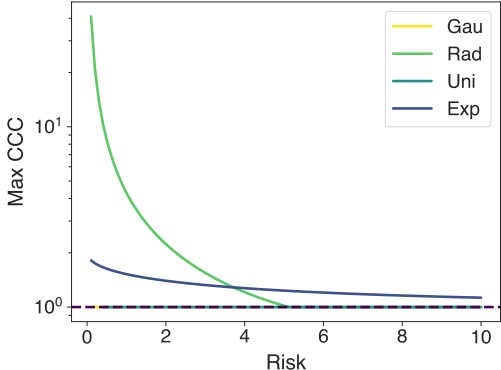

Figure 3: The maximum over $c$ of the (CCC) for various values of the risk. Notice that the maximum value of the (CCC) for both uniform and Gaussian noise is 1, corresponding to unclipped SGD. Plots are computed with $\sigma = 7$, $p = 0.5$ where $p$ is a parameter in the Rademacher-like noise.

The following theorems give concrete examples of our results and apply in both the isotropic and anisotropic setting. We show that the (CCC) cannot be satisfied when the data are Gaussian. Then, we show that, as before, broadly distributed gradients can benefit from clipping (even with all finite moments). Both proofs straightforwardly apply the definitions of $\mu_c$ and $\nu_c$ (Appendix F).

**Theorem 5.** *If $\mathbf{x} \sim N(0, \mathbf{K})$ and $\epsilon \sim N(0, \sigma^2)$ then $\mu_c^2(R)/\nu_c(R) \le 1$ for all $R, c > 0$.*

Hence, in this case clipped SGD never improves over unclipped SGD (in the sense of Theorem 4).

**Theorem 6.** *Consider $\mathbf{x} \sim N(0, \mathbf{K})$ and the Rademacher-like noise as described in Theorem 3. Then, there is an $r > 0$ depending on $p$ and $\lambda$ so that when $R_t \le r$ there always exists $c(t)$ such that the (CCC) is satisfied.*

It is interesting to note that the (CSC) is automatically satisfied if the (CCC) is, implying that when gradient clipping improves SGD's performance, it also enhances its stability. This dual benefit suggests that in some settings clipping can be used to achieve both efficient and stable training.

### 5.3 A HEURISTIC FOR THE OPTIMAL CLIPPING SCHEDULE

When we consider the max-(CCC) clipping schedule in light of our simplifications of $\mu_c$ and $\nu_c$ (Equation (10a)) we can arrive at a simple heuristic for the max-(CCC) optimal clipping schedule itself. Using Equation (10a), we see that

$$\frac{\kappa_l^2}{\kappa_u} \le \frac{\mu_c^2(R_t)}{\nu_c(R_t)} \le \frac{\kappa_u^2}{\kappa_l}. \tag{24}$$

This suggests that the upper bound might be attained by using the clipping schedule where $c(t)$ is proportional to $\sqrt{2R_t + \sigma^2}$ along with the approximate compensated learning rate

$$c^a(t) = \kappa\sqrt{2R_t + \sigma^2} \qquad\qquad \eta^a(t,c) = \eta(t)\max\left(1, 1/\kappa\right), \tag{25}$$

where $\kappa \in \mathbb{R}^+$ is a *single* scalar which can be tuned. In Figure 4, we compare: unclipped SGD, clipped SGD using the max-(CCC) schedule and compensated learning rate, clipped SGD the heuristic schedules from Equation (25). To obtain the max-(CCC) schedule we numerically optimize at each step in solving the ODE equation. In contrast, we tune only $\kappa$ to obtain our heuristic schedules.

As expected, we see no improvement by clipping for Gaussian noise (Figure 4a). In contrast, with Rademacher-like noise, clipping allows SGD to learn faster and reach a lower value of the risk (Figure 4b). Remarkably, the heuristic schedule obtains essentially identical performance as the optimal schedule. We show that this heuristic is effective on Wikitext2 data for next-token prediction and explore stability to hyper-parameter changes in Appendix C.4. This heuristic shows promising potential to improve the training of clipped SGD with minimal tuning effort, however, a comprehensive evaluation is left for future work as it warrants a dedicated study to assess its effectiveness and generalizability.

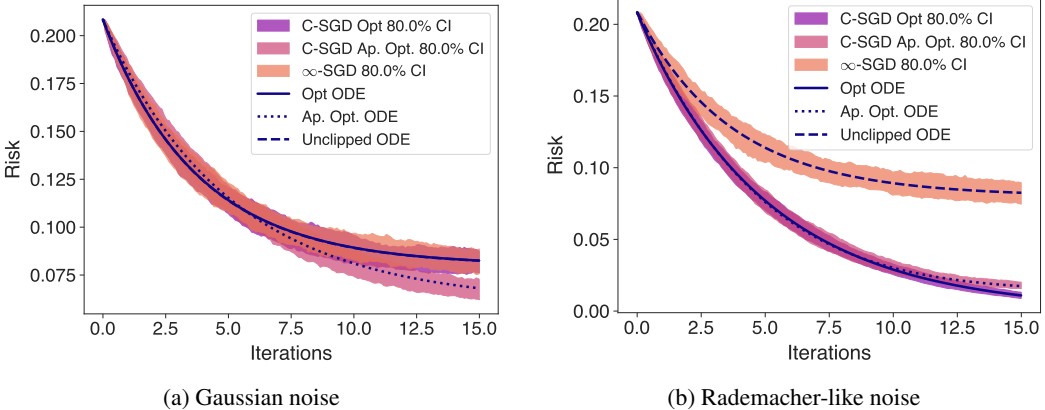

(a) Gaussian noise                    (b) Rademacher-like noise

Figure 4: Results of clipped versus unclipped SGD under the setting of Theorem 4. We compare the optimal max-(CCC) to the heuristic schedule in Equation (25). Notice that clipping cannot improve SGD in the setting with Gaussian noise while it noticeably improves performance with Rademacher-like noise. Moreover, the heuristic schedule and the optimal schedule perform very similarly. The unclipped learning rate is constantly $\eta = 0.4$ while $\sigma = 0.8$. We compare Gaussian and Rademacher-like noise with $p = 0.2$. SGD is presented with 80% confidence intervals over 100 runs.

## 6  CONCLUSION

Our analysis of high-dimensional streaming settings shows that the effectiveness of clipping hinges on two key quantities: descent and variance reduction factors $\mu_c$ and $\nu_c$. The structure of the noise, model, and data then determine the dynamics of $\mu_c$ and $\nu_c$ for a given clipping threshold. This allows us to compare clipped SGD to unclipped SGD with learning rate and clipping schedules. Clipping can be beneficial in the setting of non-Gaussian noise; in certain noisy regimes, clipping helps filter noisy datapoints more than non-noisy ones. The key is that the gradient norm becomes a strong-enough proxy for the "quality" of a datapoint, and can be used to effectively filter each point.

The local stability of clipped SGD depends on the ratio of $\mu_c$ to $\nu_c$, the (CSC). The maximum stable learning rate can be increased by clipping if clipping reduces the average square gradient norm more than the probability of clipping. This can be achieved for broad distributions of gradients. Similarly, clipping improves optimization if the ratio of $\mu_c^2$ to $\nu_c$ exceeds 1, the (CCC). This quantity informs when the tradeoff between biasing training against the gradient and reducing the variance pays off.

One future direction is to consider more complex models and losses. Exact risk curves have been derived in the unclipped SGD setting on more general losses (Collins-Woodfin et al., 2023); some of these results are likely adaptable to the clipped SGD setting. Additionally, important quantities from the analysis of high-dimensional linear models can be measured in real networks (via linearization) and can be used to analyze learning dynamics (Agarwala & Pennington, 2024). We believe that the generalized versions of $\mu_c$ and $\nu_c$ will be interesting to study in real networks.

ACKNOWLEDGEMENTS

We thank Lechao Xiao for his feedback on the clarity and accuracy of our work, which significantly improved the manuscript. We also extend our gratitude to Courtney Paquette for her proofreading and assistance in shaping the final draft.

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

## A   FULL FORMULATION OF THEOREM 1 WITH NON-GAUSSIAN DATA

To state the more general version of Theorem 1, we require some additional technical assumptions. Along with all of the assumptions described in Section 2 we will further assume:

**Assumption 4.** *For some constant $\upsilon \geq 1$ and any fixed $\boldsymbol{\theta}$ with $\|\boldsymbol{\theta}\| \leq 1$, we have $\|\mathbf{x}^T\boldsymbol{\theta}\|_{\psi_2} \leq \upsilon$ and the data satisfy a Hanson-Wright inequality: for all $t \geq 0$ and any fixed matrix $\boldsymbol{B}$,*

$$\mathbb{P}(|\mathbf{x}^T\boldsymbol{B}\mathbf{x} - \mathbb{E}[\mathbf{x}^T\boldsymbol{B}\mathbf{x}]| \geq t) \leq 2\exp\left(-\min\left\{\frac{t^2}{\upsilon^4\|\sqrt{\mathbf{K}}B\sqrt{\mathbf{K}}\|_F^2}, \frac{t}{\upsilon^2\|\sqrt{\mathbf{K}}B\sqrt{\mathbf{K}}\|}\right\}\right), \quad (26)$$

*where $\mathbf{K}$ is the covariance of the data.*

**Assumption 5.** *$\mu$ and $\nu$ satisfy the following Lipschitz-like bounds for some constants $C_\mu$ and $C_\nu$.*

$$|\mu(x) - \mu(y)| \leq C_\mu \frac{|\mathcal{R}(x) - \mathcal{R}(y)|}{\min_{z\in\{x,y\}}\mathcal{R}(z)} \quad (27)$$

$$|\nu(x)\mathcal{P}(x) - \nu(y)\mathcal{P}(y)| \leq C_\nu\left(1 + \frac{\sigma}{\sqrt{\mathcal{R}(x) + \mathcal{R}(y)}}\right)|\mathcal{R}(x) - \mathcal{R}(y)| \quad (28)$$

*where $\mathcal{R}$ is the risk.*

We may now state our more general version of Theorem 1. Here we use $\bar{c} = \max_t c(t)$ and $\bar{\eta} = \max_t \eta(t)$.

**Theorem 7.** *There is a constant $\mathcal{C} = \mathcal{C}(\upsilon, (n/d), c, \eta, (1 + \|V_0\|^2))$ and a constant $c = c(\upsilon)$ so that for any $1 \leq u \leq cd$*

$$\sup_{0\leq k\leq n}\left\|\begin{bmatrix}\mathcal{R}(\boldsymbol{\theta}_k)\\ \mathcal{D}(\boldsymbol{\theta}_k)\end{bmatrix} - \begin{bmatrix}\mathcal{R}(\boldsymbol{\Theta}_{k/d})\\ \mathcal{D}(\boldsymbol{\Theta}_{k/d})\end{bmatrix}\right\| \leq \mathcal{C}\exp\left(\int_0^{n/d}\frac{C_\nu\eta^2(s)\sigma\,\mathrm{d}s}{\sqrt{\mathcal{R}(\boldsymbol{\Theta}_s) + \mathcal{R}(\boldsymbol{\theta}_{sd})}}\right)u\log(d)d^{-1/2}, \quad (29)$$

*with probability at least $1 - e^{-u}$ and provided the right hand side is less than 1. The coefficient $\mathcal{C}$ can be bounded by*

$$\mathcal{C} \leq C\sqrt{n/d}\bar{\eta}\upsilon^2((1 + \|V_0\|^2)\upsilon^2 + c^2\sqrt{n/d})\exp\left(C\times(1 + C_\mu)\max\{\bar{\eta}, \bar{\eta}^2\}(n/d)\right)$$

*for an absolute constant $C > 0$.*

We note that when $\sigma = 0$ (so there is no noise) we arrive at the simpler conclusion that

$$\sup_{0\leq k\leq n}\left\|\begin{bmatrix}\mathcal{R}(\boldsymbol{\theta}_k)\\ \mathcal{D}(\boldsymbol{\theta}_k)\end{bmatrix} - \begin{bmatrix}\mathcal{R}(\boldsymbol{\Theta}_{k/d})\\ \mathcal{D}(\boldsymbol{\Theta}_{k/d})\end{bmatrix}\right\| = \mathcal{O}(d^{-1/2}).$$

We note also that if $\eta(s) \equiv \eta$, the risk will be bounded below by a constant that depends only on $\eta, c, \sigma$ with high probability (and provided there is no warm start), and hence again this coefficient can be bounded with high probability in a similar way to $\mathcal{C}$. Moreover, for any desired $d$-independent risk threshold $R_0$, if one makes $d$ sufficiently large, then with very high probability, two risk curves will agree up to the point they cross below this risk threshold.

# B  PROOF OF MAIN THEOREMS

In this section we prove both Theorem 1 and Theorem 7.

## B.1  PROOF OF THEOREM 1

In order to prove the version of our Theorem with Gaussian data, it suffices to check that Gaussians satisfy both Assumption 4 and 5.

It is a standard fact that the Hanson-Wright inequality is satisfied for Gaussians Vershynin (2018).

Let $z$ be standard Gaussian then recall from (9),

$$\mu_c(\boldsymbol{\theta}) = \mathbb{P}(|\langle \mathbf{x}, \boldsymbol{\theta} - \boldsymbol{\theta}^* \rangle| \leq c) \tag{30}$$

$$= \mathbb{P}(|\sqrt{2\mathcal{R}(\boldsymbol{\theta})} z - \epsilon| \leq c). \tag{31}$$

With a slight abuse of notation, we condition on $\epsilon$ and use $I = (\epsilon - c, \epsilon + c)$ to express

$$|\mu_c(\boldsymbol{\theta}_1) - \mu_c(\boldsymbol{\theta}_2)| = \left| \mathbb{P}\left( z \in \frac{I}{\sqrt{2\mathcal{R}(\boldsymbol{\theta}_1)}} \middle| \epsilon \right) - \mathbb{P}\left( z \in \frac{I}{\sqrt{2\mathcal{R}(\boldsymbol{\theta}_2)}} \middle| \epsilon \right) \right| \tag{32}$$

$$\leq \left| \mathbb{P}\left( z \leq \frac{c + \epsilon}{\sqrt{\mathcal{R}(\boldsymbol{\theta}_1)}} \middle| \epsilon \right) - \mathbb{P}\left( z \leq \frac{c + \epsilon}{\sqrt{\mathcal{R}(\boldsymbol{\theta}_2)}} \middle| \epsilon \right) \right| \tag{33}$$

$$+ \left| \mathbb{P}\left( z \leq \frac{c - \epsilon}{\sqrt{\mathcal{R}(\boldsymbol{\theta}_1)}} \middle| \epsilon \right) - \mathbb{P}\left( z \leq \frac{c - \epsilon}{\sqrt{\mathcal{R}(\boldsymbol{\theta}_2)}} \middle| \epsilon \right) \right|. \tag{34}$$

Without loss of generality, we assume $c + \epsilon/\mathcal{R}(\boldsymbol{\theta}_2) \leq c + \epsilon/\mathcal{R}(\boldsymbol{\theta}_1)$, then the former term may be bounded by

$$\left| \mathbb{P}\left( z \leq \frac{c + \epsilon}{\sqrt{\mathcal{R}(\boldsymbol{\theta}_1)}} \middle| \epsilon \right) - \mathbb{P}\left( z \leq \frac{c + \epsilon}{\sqrt{\mathcal{R}(\boldsymbol{\theta}_2)}} \middle| \epsilon \right) \right| \tag{35}$$

$$\leq \frac{|c + \epsilon|}{2\sqrt{\pi}} e^{-\frac{(c+\epsilon)^2}{4\mathcal{R}(\boldsymbol{\theta}_1)}} \left| \frac{\sqrt{\mathcal{R}(\boldsymbol{\theta}_1)} - \sqrt{\mathcal{R}(\boldsymbol{\theta}_2)}}{\sqrt{\mathcal{R}(\boldsymbol{\theta}_1)\mathcal{R}(\boldsymbol{\theta}_2)}} \right|. \tag{36}$$

Maximizing $te^{-t^2/4\mathcal{R}(\boldsymbol{\theta}_1)}$ in $t$ yields

$$\left| \mathbb{P}\left( z \leq \frac{c + \epsilon}{\sqrt{\mathcal{R}(\boldsymbol{\theta}_1)}} \middle| \epsilon \right) - \mathbb{P}\left( z \leq \frac{c + \epsilon}{\sqrt{\mathcal{R}(\boldsymbol{\theta}_2)}} \middle| \epsilon \right) \right| \leq \frac{|\mathcal{R}(\boldsymbol{\theta}_1) - \mathcal{R}(\boldsymbol{\theta}_2)|}{\sqrt{2\pi \mathcal{R}(\boldsymbol{\theta}_2)}(\sqrt{\mathcal{R}(\boldsymbol{\theta}_1)} + \sqrt{\mathcal{R}(\boldsymbol{\theta}_2)})} \tag{37}$$

$$\leq C_\mu \frac{|\mathcal{R}(\boldsymbol{\theta}_1) - \mathcal{R}(\boldsymbol{\theta}_2)|}{\min_{z \in \{\boldsymbol{\theta}_1, \boldsymbol{\theta}_2\}} \mathcal{R}(z)}. \tag{38}$$

By applying the same argument to the latter term of (34), we see that

$$|\mu(\boldsymbol{\theta}_1) - \mu(\boldsymbol{\theta}_2)| \leq C_\mu \frac{|\mathcal{R}(\boldsymbol{\theta}_1) - \mathcal{R}(\boldsymbol{\theta}_2)|}{\min_{z \in \{\boldsymbol{\theta}_1, \boldsymbol{\theta}_2\}} \mathcal{R}(z)}, \tag{39}$$

as desired.

To show (28), let us first first define $f(\xi, \epsilon) = \mathbb{E}\left[\mathrm{clip}_c^2\left(\xi z - \epsilon\right)\Big|\,\epsilon\right]$. Upon conditioning on $\epsilon$, it follows that $\nu(\boldsymbol{\theta})\mathcal{P}(\boldsymbol{\theta}) = f\left(\sqrt{2\mathcal{R}(x)}, \epsilon\right)$. Differentiating with respect to $\xi$, we see that

$$\frac{\partial}{\partial \xi} f(\xi) = \mathbb{E}\left[2z\,\mathrm{clip}_c(\xi z - \epsilon)\mathbb{1}_{|\xi z - \epsilon| \leq c}\right] \tag{40}$$

$$= \frac{2}{\sqrt{2\pi}} \int_{(-c+\epsilon)/\xi}^{(c+\epsilon)/\xi} (\xi z - \epsilon)z e^{-z^2/2}\, dz \tag{41}$$

$$= \frac{-2}{\sqrt{2\pi}} \left[(\xi z - \epsilon)e^{-z^2/2}\Big|_{z=(-c+\epsilon)/\xi}^{z=(c+\epsilon)/\xi}\right] + 2\xi\mathbb{P}\left(|\xi z - \epsilon| \leq c\,|\,\epsilon\right) \tag{42}$$

$$= \frac{-2}{\sqrt{2\pi}} \left[(c - \epsilon + \epsilon)e^{-(c-\epsilon)^2/2\xi^2} + (c + \epsilon - \epsilon)e^{-(c+\epsilon)^2/2\xi^2}\right] \tag{43}$$

$$+ 2\xi\mathbb{P}\left(|\xi z - \epsilon| \leq c\,|\,\epsilon\right). \tag{44}$$

Upon noting that $\xi\left(\frac{c+\epsilon}{\xi\sqrt{2\pi}}e^{-\frac{(c+\epsilon)^2}{2\xi^2}}\right) \leq \xi C$, for some absolute constant $C > 0$ we may bound the absolute value of $\frac{\partial}{\partial \xi} f(\xi, \epsilon)$ by

$$\left|\frac{\partial}{\partial \xi} f(\xi, \epsilon)\right| \leq 4C\xi + 2|\epsilon|. \tag{45}$$

Hence, without loss of generality, if we assume that $\sqrt{2\mathcal{R}(\boldsymbol{\theta}_1)} \leq \sqrt{2\mathcal{R}(\boldsymbol{\theta}_2)}$ and conditioning on $\epsilon$, we obtain

$$|\nu(\boldsymbol{\theta}_1)\mathcal{P}(\boldsymbol{\theta}_1) - \nu(\boldsymbol{\theta}_2)\mathcal{P}(\boldsymbol{\theta}_2)| \leq \int_{\sqrt{2\mathcal{R}(\boldsymbol{\theta}_1)}}^{\sqrt{2\mathcal{R}(x_2)}} \left|\frac{\partial}{\partial \xi} f(\xi, \epsilon)\right| d\xi \tag{46}$$

$$\leq 2c\,|\mathcal{R}(\boldsymbol{\theta}_1) - \mathcal{R}(\boldsymbol{\theta}_2)| + 2\mathbb{E}|\epsilon|\left|\sqrt{2\mathcal{R}(\boldsymbol{\theta}_1)} - \sqrt{2\mathcal{R}(\boldsymbol{\theta}_2)}\right|, \tag{47}$$

which completes the proof of (28).

## B.2 Proof of Theorem 7

We now prove the general version of our main result.

We simplify notation by studying the iterations $v_k = \boldsymbol{\theta}_k - \boldsymbol{\theta}^*$. We shall also write $\tilde{\eta}_k = \eta(k/d)$ so that $\tilde{\eta}_k/d = \eta_k$. Before proving theorem 7 we first show a series of lemmas following closely the proof techniques of Collins-Woodfin & Paquette (2023).

**Notation 1.** *It is helpful to formulate some results in terms of tensor products. We use $x \otimes y$ to refer the tensor product of $x$ and $y$.*

**Notation 2.** *We use $C$ to refer to a generic constant which may change from line to line.*

With a slight abuse of notation, extend $\{v_k\}_{k\geq 0}$ to be indexed by continuous time $t \in \mathbb{R}^+$ by $v_t = v_{\lfloor t \rfloor}$. Let $q$ be a quadratic. Via its Taylor expansion, we may write the updates of $q$ by

$$q(v_{k+1}) - q(v_k) = -\frac{\tilde{\eta}_k}{d}\nabla q(v_k)^T \mathrm{clip}_{c\sqrt{d}}(\ell_k \mathbf{x}_{k+1}) + \frac{\tilde{\eta}_k^2}{2d^2}\left\langle \nabla^2 q, (\mathrm{clip}_{c\sqrt{d}}(\ell_k \mathbf{x}_{k+1}))^{\otimes 2}\right\rangle. \tag{48}$$

This update can be decomposed into errors, martingale parts and predictable parts

$$q(v_{k+1}) - q(v_k) = -\frac{\tilde{\eta}_k}{d}\mu(v_k)\nabla q(v_k)^T \mathbf{K} v_k + \frac{\tilde{\eta}_k^2}{d^2}\nu(v_k)\mathcal{P}(v_k)\,\mathrm{Tr}(\nabla^2 q\mathbf{K}) \tag{49}$$
$$+ \Delta\mathcal{M}_k^{lin} + \Delta\mathcal{M}_k^{quad} + \Delta E_k.$$

Where we have martingale and error increments being contributed from both the linear and quadratic terms. The specific form of these terms may be seen in section B.3. We will relate these quadratics to

a manifold of functions which will close under the gradient and Hessian operations above. Choose this family of functions to be

$$Q = \{v \mapsto v^T R(z; \mathbf{K})v, \quad \forall z \in \Omega\} \tag{50}$$

where $\Omega$ is a circle of radius 2 and thus enclosing the eigenvalues of $\mathbf{K}$. We further define the stopping time $\tau$ for a parameter $M$.

$$\tau = \inf\{k : \|v_k\| \geq M\} \cup \{td : \|V_t\| \geq M\}, \tag{51}$$

and the stopped processes,

$$v_k^\tau = v_{k \wedge \tau} \qquad\qquad V_t^\tau = V_{t \wedge (\tau/d)}. \tag{52}$$

We will first prove Theorem 7 for the stopped process $\{v_k^\tau\}_{k \geq 0}$ and $\{V_t^\tau\}_{t \geq 0}$ and then bound the probability that $\tau \leq n$.

**Lemma 1.** *There is an absolute constant $C > 0$ so that*

$$\sup_{0 \leq t \leq n/d} |q(v_{td}^\tau) - q(V_t^\tau)| \leq \sup_{0 \leq t \leq n/d} \left( |\mathcal{M}_{td}^{\tau, lin}| + |\mathcal{M}_{td}^{\tau, quad}| + |E_{td}^\tau| + |\mathcal{M}_t^{\tau, SDE}| \right) \tag{53}$$

$$+ \int_0^{n/d} \left( C \max\{\bar{\eta}, \bar{\eta}^2\} + \frac{C_\nu \eta^2(s)(\mathfrak{m}_s + \sigma)}{\mathfrak{m}_s} + 2C_\mu \bar{\eta} \right) \sup_{q \in Q} |q(v_{sd}^\tau) - q(V_s^\tau)| \, \mathrm{d}s, \tag{54}$$

where $\mathfrak{m}_s$ is sum of risks $\mathfrak{m}_s = \sqrt{\mathcal{R}(\boldsymbol{\theta}_{sd}^\tau) + \mathcal{R}(\boldsymbol{\Theta}_{sd}^\tau)}$.

*Proof.* When context is clear, we will write $R(z; \mathbf{K}) = R(z)$. We begin by noting that for all $q \in Q$ since for eigenvalues and eigenvectors $(\lambda_i, \omega_i)$ of $\mathbf{K}$,

$$\left| \nabla q(v)^T \mathbf{K}v \right| = \left| \sum_i \frac{\lambda_i}{(\lambda_i - z)} \langle v, \omega_i \rangle^2 \right| \leq \sum_i \lambda_i \langle v, \omega_i \rangle^2 = \langle \mathbf{K}, v^{\otimes 2} \rangle.$$

The same bound holds for the gradient term, and we conclude that for all $q \in Q$

$$|\nabla q(v)^T \mathbf{K}v| \leq 2\mathcal{R}(v + \boldsymbol{\theta}^*).$$

Given a $g \in Q$, by (49), we obtain

$$g(v_t^\tau) = g(v_0^\tau) - \int_0^t \frac{\eta(s)}{d} \mu(v_s^\tau) \nabla g(v_s^\tau)^T \mathbf{K}v_s^\tau \mathrm{d}s + \int_0^t \frac{\eta^2(s)}{d^2} \nu(v_s^\tau) \mathcal{P}(v_s^\tau) \operatorname{Tr}(\mathbf{K}\nabla^2 g) \mathrm{d}s \tag{55}$$

$$+ \mathcal{M}_t^{\tau, lin} + \mathcal{M}_t^{\tau, quad} + E_t^\tau.$$

Similarly, by Itô's lemma

$$g(V_t^\tau) = g(V_0^\tau) - \int_0^t \eta(s) \mu(V_s^\tau) \nabla g(V_s^\tau)^T \mathbf{K}V_s^\tau \mathrm{d}s$$
$$+ \int_0^t \frac{\eta^2(s)}{d^2} \nu(V_s^\tau) \mathcal{P}(V_s^\tau) \operatorname{Tr}(\mathbf{K}\nabla^2 g) \mathrm{d}s + \mathcal{M}_t^{\tau, SDE}, \tag{56}$$

where

$$\mathcal{M}_t^{\tau, SDE} = \int_0^t \frac{\eta(s)}{\sqrt{d}} \nabla g(V_s^\tau)^T \sqrt{2\mathbf{K}\nu(V_s^\tau)\mathcal{P}(V_s^\tau)} dB_s. \tag{57}$$

First, we will show that for any $g \in Q$ and any $x_1, x_2 \in \mathbb{R}^d$

$$|\nabla g(x_1)^T \mathbf{K} x_1 - \nabla g(x_2)^T \mathbf{K} x_2| \leq 4 \sup_{g \in Q} |g(x_1) - g(x_2)|. \tag{58}$$

The statement is obvious if $g(x) = q(x)$. If $g(x) = \nabla q(x)^T R(z) x$ then $\nabla g(x) = \nabla^2 q R(z) x + R(z) \nabla q(x)$ and using Cauchy's integral formula we can see,

$$\nabla g(x)^T \mathbf{K} x = x^T R(z) \nabla^2 q \mathbf{K} x + \nabla q(x)^T R(z) \mathbf{K} x \tag{59}$$

$$= \underbrace{-\frac{1}{2\pi i} \oint_\Omega y x^T R(z) \nabla^2 q R(y) x \, dy}_{T_1} \underbrace{- \frac{1}{2\pi i} \oint_\Omega \nabla q(x)^T R(z) x \, dz}_{T_2} + \underbrace{z \nabla q(x)^T R(z) x}_{T_3}. $$

$$\tag{60}$$

For any $z$ on $\Omega$ we have that $\|R(z)\|_{op} \leq 1$. Furthermore, the arc-length of $\Omega$ is $8\pi$. Therefore, we have

$$|T_1(x_1) - T_1(x_2)| \leq \frac{1}{2\pi} \oint_\Omega |y| \left| x_1^T R(z) \nabla^2 q R(y) x_1 - x_2^T R(z) \nabla^2 q R(y) x_2 \right| dy \tag{61}$$

$$\leq 8 \sup_{g \in Q} |g(x_1) - g(x_2)|, \tag{62}$$

and

$$|T_2(x_1) - T_2(x_2)| \leq \frac{1}{2\pi} \oint_\Omega |\nabla q(x_1)^T R(z) x_1 - \nabla q(x_2)^T R(z) x_2| dz \tag{63}$$

$$\leq 4 \sup_{g \in Q} |g(x_1) - g(x_2)|. \tag{64}$$

If $g(x) = x^T R(z) \nabla^2 q R(y) x$, then using the identity $R(z) \mathbf{K} = \mathbf{I} + z R(z)$,

$$\nabla g(x)^T \mathbf{K} x = x^T R(y) \nabla^2 q x + z x^T R(y) \nabla^2 q R(z) x + x^T R(z) \nabla^2 q x + y x^T R(z) \nabla^2 q R(y) x. \tag{65}$$

By the same methods as above, we see

$$|\nabla g(x_1)^T \mathbf{K} x_1 - \nabla g(x_2)^T \mathbf{K} x_2| \leq 24 \sup_{g \in Q} |g(x_1) - g(x_2)|. \tag{66}$$

It is simple to account for the presence of the functions $\mu$ and $\nu$. Using Assumptions 5

$$|\nu(v_{td}^\tau) \mathcal{P}(v_{td}^\tau) - \nu(V_t^\tau) \mathcal{P}(V_{td}^\tau)| \leq \sup_{g \in Q} |g(v_{td}^\tau) - g(V_t^\tau)| \frac{C_\nu(\mathfrak{m}_t + \sigma)}{\mathfrak{m}_t}. \tag{67}$$

As for $\mu$, adding and subtracting $\mu(v_{td}^\tau) g(V_{td}^\tau)$, using $\mu \leq 1$ and $g(V_{td}^\tau) \leq 2\mathcal{R}(\Theta_{td}^\tau)$

$$|\mu(v_{td}^\tau) g(v_{td}^\tau) - \mu(V_t^\tau) g(V_t^\tau)| \leq \sup_{g \in Q} |g(v_{td}^\tau) - g(V_t^\tau)| \left( 1 + \frac{C_\mu 2\mathcal{R}(\Theta_{td}^\tau)}{\min\{\mathcal{R}(\theta_{td}^\tau), \mathcal{R}(\Theta_{td}^\tau)\}} \right). \tag{68}$$

Note we could have also added and subtracted $\mu(V_{td}^\tau) g(v_{td}^\tau)$, and so picking whichever is better, we arrive at

$$|\mu(v_{td}^\tau) g(v_{td}^\tau) - \mu(V_t^\tau) g(V_t^\tau)| \leq \sup_{g \in Q} |g(v_{td}^\tau) - g(V_t^\tau)| (1 + 2C_\mu).$$

This completes the claim.

$\square$

**Lemma 2.** *There is an absolute constant $C > 0$ so that for any quadratic $q$ with $\|q\|_{C^2} \leq 1$ any $n \leq dT$ with $T \geq 1$, any $1 \leq u \leq d$,*

$$\left| \sup_{0 \leq k \leq n} \mathcal{M}_k^{\tau,lin} \right| \leq C\sqrt{T}\overline{\eta}(2+M)^2 v^2 d^{-1/2}u, \tag{69}$$

$$\left| \sup_{0 \leq k \leq n} \mathcal{M}_t^{\tau,quad} \right| \leq C\sqrt{T}\overline{\eta}c^2 v^2 d^{-1/2}u, \tag{70}$$

$$\left| \sup_{0 \leq k \leq n} E_t^{\tau} \right| \leq CT\overline{\eta}(2+M)^2 v^4 d^{-1/2}, \tag{71}$$

$$\left| \sup_{0 \leq t \leq n/d} \mathcal{M}_t^{\tau,SDE} \right| \leq C\sqrt{T}\overline{\eta}(2+M)^2 v^2 d^{-1/2}u. \tag{72}$$

*with probability at least $1 - e^{-u}$.*

The proof of lemma 2 is deferred to appendix B.3.

**Lemma 3.** *There is an absolute constant $C > 0$ so that for any $m > 0$, there exists a $\bar{Q} \subseteq Q$ with $|\bar{Q}| \leq Cd^{2m}$ such that for all $q \in Q$, there is some $\bar{q} \in \bar{Q}$ that satisfies $\|\bar{q} - q\|_{C^2} \leq d^{-2m}$.*

*Proof.* With assumption 2, the arc length of $\Omega$ is fixed independent of $d$. Thus, we may construct $\bar{Q}$ by restricting $Q$ to a minimal $d^{-2m}$-net of $\Omega$. □

The proof of Theorem 7 now follows easily from these results. By Lemmas 1 and 2, there is an absolute constant $C$ so that for any $u \geq 1$

$$|\bar{q}(v_{td}^{\tau}) - \bar{q}(V_t^{\tau})| \leq C\sqrt{T}\overline{\eta}v^2 d^{-1/2}\left((2+M)^2 v^2 u + c^2\sqrt{T}\right) + \int_0^t \mathfrak{L}_s \max_{q \in \bar{Q}} |q(v_{sd}^{\tau}) - q(V_s^{\tau})|\mathrm{d}s, \tag{73}$$

on an event of probability at least $1 - e^{-u}$, and where we have set

$$\mathfrak{L}_s := \left(C\max\{\overline{\eta},\overline{\eta}^2\} + \frac{C_\nu \eta^2(s)(\mathfrak{m}_s + \sigma)}{\mathfrak{m}_s} + 2C_\mu\overline{\eta}\right).$$

Then, from Lemma 3 with $m = 1$ and increasing the absolute constant $C > 0$ so that for all $t \leq T$

$$\sup_{q \in Q} |q(v_{td}^{\tau}) - q(V_t^{\tau})| \leq C\sqrt{T}\overline{\eta}v^2 d^{-1/2}\left((2+M)^2 v^2 u + c^2\sqrt{T}\right) + \int_0^t \mathfrak{L}_s \max_{q \in \bar{Q}} |q(v_{sd}^{\tau}) - q(V_s^{\tau})|\mathrm{d}s, \tag{74}$$

except on an event of probability $Cd^8 e^{-u}$.

An application of Gronwall's inequality gives

$$\sup_{q \in Q} \sup_{0 \leq t \leq T} |q(v_{td}^{\tau}) - q(V_t^{\tau})| \leq C\sqrt{T}\overline{\eta}v^2 d^{-1/2}\left((2+M)^2 v^2 u + c^2\sqrt{T}\right)\exp\left(\int_0^T \mathfrak{L}_s \mathrm{d}s\right). \tag{75}$$

Now we note that by contour integration, both the risk $v \mapsto \langle \mathbf{K}, v^{\otimes 2}\rangle$ and suboptimality $v \mapsto \|v\|^2$ both can be estimated by

$$\max\{\left|\|\boldsymbol{\Theta}_{td}^{\tau} - \boldsymbol{\theta}^*\|^2 - \|\boldsymbol{\Theta}_{td}^{\tau} - \boldsymbol{\theta}^*\|^2\right|, 2\left|\mathcal{R}(\boldsymbol{\theta}_{td}^{\tau}) - \mathcal{R}(\boldsymbol{\Theta}_{td}^{\tau})\right|\} \leq 4\sup_{q \in Q}|q(v_{td}^{\tau}) - q(V_t^{\tau})|,$$

proving our claim for the stopped processes. Now, it will be shown that with overwhelming $\tau$ does not occur for $n \leq dT$. It suffices to show that following lemma.

**Lemma 4.** *There is an absolute constant $C > 0$ so that for all $r \geq 0$ and all $T \geq 0$ with probability at least $1 - 2e^{-r^2/2}$ for all $0 \leq s \leq T$,*

$$e^{-C\max\{\overline{\eta},\overline{\eta}^2\}s - C\overline{\eta}\sqrt{T}d^{-1/2}r} \leq \frac{\|V_s\|^2}{\|V_0\|^2} \leq e^{C\max\{\overline{\eta},\overline{\eta}^2\}s + C\overline{\eta}\sqrt{T}d^{-1/2}r}. \tag{76}$$

*Proof.* Consider $\varphi(V_t) = \log(1 + \|V_t\|^2)$. Then,

$$
\begin{aligned}
\mathrm{d}\varphi(V_t) = & - 2\eta(t)\frac{\mu(X_t)}{1+\|V_t\|^2}\nabla\mathcal{P}(V_t)^T V_t \mathrm{d}t + \frac{\eta^2(t)2\nu(V_t)\mathcal{P}(V_t)}{1+\|V_t\|^2}\frac{\mathrm{Tr}(\mathbf{K})}{d}\mathrm{d}t \\
& - \frac{2\eta^2(t)\nu(V_t)\mathcal{P}(V_t)}{d(1+\|V_t\|^2)^2}\langle V_t \otimes V_t, \mathbf{K}\rangle \mathrm{d}t + \frac{2\eta(t)\sqrt{2\nu(V_t)\mathcal{P}(V_t)}}{\sqrt{d}(1+\|V_t\|^2)}\left\langle V_t, \sqrt{\mathbf{K}}\mathrm{d}B_t\right\rangle.
\end{aligned}
\tag{77}
$$

Note that, $\mathrm{Tr}(\mathbf{K})/d = \|\mathbf{K}\| = 1$ so the drift terms are all bounded above and below by absolute constants multiplied by $\max\{\overline{\eta}, \overline{\eta}^2\}$. Meanwhile, the quadratic variation is bounded by

$$
\langle\varphi(V)\rangle_t = \int_0^t \frac{8\eta^2(s)}{d}\nu(V_s)\mathcal{P}(V_s)\frac{\langle V_s \otimes V_s, \mathbf{K}\rangle}{(1+\|V_s\|^2)^2}\mathrm{d}s
\tag{78}
$$

$$
\leq 8C\frac{\overline{\eta}^2}{d}t,
\tag{79}
$$

for $C$ an absolute constant.

And so, for all $r \geq 0$, setting $f(t)$ to be the integrated drift terms from (77)

$$
\mathbb{P}(\max_{1\leq t\leq T}|\varphi(V_t) - f(t)| \geq C\overline{\eta}\sqrt{T}/\sqrt{d}r) \leq 2\exp(-r^2/2).
\tag{80}
$$

This implies the claim immediately as $|f(t)| \leq C\max\{\overline{\eta}, \overline{\eta}^2\}t$ for all $t$. $\qquad\square$

We can now conclude the main theorem, noting that if for some fixed $T$, if we pick $\tilde{M}$ so that

$$
(2+\tilde{M})^2 = \max\left\{C(1+\|V_0\|^2)\exp\left(\int_0^T Cs\right), (2+2)^2\right\}
$$

then with probability at least $1 - e^{-d}$, $\|V_t\|$ remains below $\tilde{M}$ up time $T$. As single steps of clipped SGD cannot increase the norm of $v_k$ by more than a factor of 2 (with probability at least $1 - e^{-cd}$), we conclude that if $\tau \leq Td$, using (75)

$$
M^2 = \|v_\tau\|^2 \leq 4(\tilde{M})^2 + 4C\sqrt{T}\overline{\eta}v^2 d^{-1/2}\left((2+M)^2v^2u + c^2\sqrt{T}\right)\exp\left(\int_0^T \mathfrak{L}_s \mathrm{d}s\right).
$$

Provided $M \geq 2$ and provided that

$$
4C\sqrt{T}\overline{\eta}v^2\left(v^2u + c^2\sqrt{T}\right)\exp\left(\int_0^T \mathfrak{L}_s \mathrm{d}s\right)d^{-1/2} \leq \frac{1}{8},
\tag{81}
$$

we have

$$
M^2 \leq 4(\tilde{M})^2 + \frac{1}{2}M^2,
$$

hence we conclude that

$$
M \leq \sqrt{8}\tilde{M}.
$$

So if we pick $M$ larger than $\sqrt{8}\tilde{M}$ (which is larger than 2 by how $\tilde{M}$ was picked) we conclude that $\tau > Td$.

### B.3 BOUNDING MARTINGALES AND ERRORS

**Lemma 5.** *Martingale Bernstein inequality For $\{M_k\}_{k=0}^N$ a martingale, we define*

$$
\sigma_{k,p} = \inf\{t > 0 : \mathbb{E}[\exp(|M_k - M_{k-1}|^p/t^p)|\mathscr{F}_{k-1}] \leq 2\},
\tag{82}
$$

*then there exists an absolute constant $C > 0$ such that for all $t > 0$*

$$
\mathbb{P}\left(\sup_{1\leq k\leq N}|M_k - \mathbb{E}[M_0]| > t\right) \leq 2\exp\left(-\min\left\{\frac{t}{C\max\sigma_{k,1}}, \frac{t^2}{C\sum_{i=1}^N \sigma_{i,1}^2}\right\}\right).
\tag{83}
$$

This section is dedicated to bounding the martingale and error terms present in Equations (55) and (56). These terms are

$$\Delta \mathcal{M}_{k+1}^{\tau,lin} := -\frac{\tilde{\eta}_k}{d} \nabla q(v_k^\tau)^T \operatorname{clip}_{c\sqrt{d}}(\ell_k \mathbf{x}_{k+1}) + \frac{\tilde{\eta}_k}{d} \nabla q(v_k^\tau)^T \mathbb{E}[\operatorname{clip}_{c\sqrt{d}}(\ell_k \mathbf{x}_{k+1})|\mathscr{F}_k] \tag{84}$$

$$= -\frac{\tilde{\eta}_k}{d} \nabla q(v_k^\tau)^T \operatorname{clip}_{c\sqrt{d}}(\ell_k \mathbf{x}_{k+1}) + \frac{\tilde{\eta}_k}{d} \nabla q(v_k^\tau)^T \mathbf{K} v_k^\tau \mu(v_k^\tau) - \Delta E_k^{\tau,lin}, \tag{85}$$

and

$$\Delta \mathcal{M}_{k+1}^{\tau,quad} := \frac{\tilde{\eta}_k}{2d^2} \left\langle \nabla^2 q, \operatorname{clip}_{c\sqrt{d}}(\ell_k \mathbf{x}_{k+1})^{\otimes 2} \right\rangle - \frac{\tilde{\eta}_k}{2d^2} \left\langle \nabla^2 q, \mathbb{E}\left[\operatorname{clip}_{c\sqrt{d}}(\ell_k \mathbf{x}_{k+1})^{\otimes 2}\right] \right\rangle \tag{86}$$

$$= \frac{\tilde{\eta}_k}{2d^2} \left\langle \nabla^2 q, \operatorname{clip}_{c\sqrt{d}}(\ell_k \mathbf{x}_{k+1})^{\otimes 2} \right\rangle - \frac{\tilde{\eta}_k}{d^2} \left\langle \mathbf{K}, \nabla^2 q \right\rangle \nu(v_k)\mathcal{P}(v_k) - \Delta E_k^{\tau,quad}, \tag{87}$$

where we recall $\ell_k = \langle \mathbf{x}_{k+1}, v_k^\tau \rangle - \epsilon_{k+1}$. The error increment has contributions from both the linear—in $\tilde{\eta}_k$—and quadratic terms. More precisely,

$$\Delta E_k^\tau = \Delta E_k^{\tau,lin} + \Delta E_k^{\tau,quad}, \tag{88}$$

where

$$\Delta E_k^{\tau,lin} = -\frac{\tilde{\eta}_k}{d} \nabla q(\boldsymbol{\theta}_k^\tau)^T \mathbb{E}[\operatorname{clip}_{c\sqrt{d}}(\mathbf{x}_{k+1}\ell_k) - \mathbf{x}_{k+1} \operatorname{clip}_c(\ell_k)]$$

and

$$\Delta E_k^{\tau,quad} = \frac{\tilde{\eta}_k}{2d^2} \left( \mathbb{E}\left[ \left\langle \nabla^2 q, \mathbf{x}_{k+1}^{\otimes 2} \right\rangle \ell_k^2 \mathbb{1}_{\|\ell_k \mathbf{x}_{k+1}\|^2 \leq c^2 d} \right] - \left\langle \nabla^2 q, \mathbf{K} \right\rangle \mathbb{E}\left[ \ell_k^2 \mathbb{1}_{\ell_k^2 \operatorname{Tr}(\mathbf{K}) \leq c^2 d} \right] \right).$$

## B.4 MARTINGALE FOR THE LINEAR TERMS

We'll begin the proof for the linear terms in the increments. First, note that using the $\|q\|_{C^2}$ norm we can bound

$$\|\nabla q(x)\| \leq \|\nabla^2 q\|\|x\| + \|\nabla q(0)\| \leq \|q\|_{C^2}(1 + \|x\|). \tag{89}$$

Thus,

$$|\nabla q(v_k^\tau)^T \mathbf{K} v_k^\tau \mu(v_k^\tau)| \leq (1 + M). \tag{90}$$

From Equation (134) in the following section, for an absolute constant $C > 0$

$$|\Delta E_k^{\tau,lin}| \leq C(2 + M)^2 \overline{\eta} d^{-3/2} \mathfrak{v}^3. \tag{91}$$

Meanwhile, we can get subexponential bounds for the former terms of (85),

$$-\frac{\tilde{\eta}_k}{d} \nabla q(v_k^\tau)^T \operatorname{clip}_{c\sqrt{d}}(\ell_k \mathbf{x}_{k+1}) = -\frac{\tilde{\eta}_k}{d} \nabla q(v_k^\tau)^T \mathbf{x}_{k+1}\ell_k \mathbb{1}_{\|\ell_k \mathbf{x}_{k+1}\| \leq c\sqrt{d}} \tag{92}$$

$$-\frac{c\tilde{\eta}_k}{\sqrt{d}} \nabla q(v_k^\tau)^T \frac{\ell_k \mathbf{x}_{k+1}}{\|\ell_k \mathbf{x}_{k+1}\|} \mathbb{1}_{\|\ell_k \mathbf{x}_{k+1}\| > c\sqrt{d}}. \tag{93}$$

So, by Assumptions 1 and 4, as well as Equation (89), we have

$$\left\| \nabla q(v_k^\tau)^T \mathbf{x}_{k+1}\ell_k \mathbb{1}_{\|\ell_k \mathbf{x}_{k+1}\| < c\sqrt{d}} \right\|_{\psi_1} \leq \|\nabla q(v_t^\tau)^T \mathbf{x}_{k+1}\ell_k\|_{\psi_1} \tag{94}$$

$$\leq \|\nabla q(v_t^\tau)^T \mathbf{x}_{k+1}\|_{\psi_2}\|\ell_k\|_{\psi_2} \tag{95}$$

$$\leq (1 + M)\mathfrak{v} \times (2 + M)\mathfrak{v}. \tag{96}$$

Likewise,

$$\left\| c\sqrt{d}\nabla q(v_k^\tau)^T \frac{\ell_k \mathbf{x}_{k+1}}{\|\ell_k \mathbf{x}_{k+1}\|} \mathbb{1}_{\|\ell_k \mathbf{x}_{k+1}\| > c\sqrt{d}} \right\|_{\psi_1} \leq \left\| \nabla q(v_k^\tau)^T \ell_k \mathbf{x}_{k+1} \mathbb{1}_{\|\ell_k \mathbf{x}_{k+1}\| > c\sqrt{d}} \right\|_{\psi_1} \tag{97}$$

$$\leq \left\| \nabla q(v_k^\tau)^T \ell_k \mathbf{x}_{k+1} \right\|_{\psi_1} \tag{98}$$

$$\leq (2+M)^2 v^2. \tag{99}$$

Thus, for some absolute constant $C > 0$

$$\sigma_{k,1} = \inf\{t > 0 : \mathbb{E}[\exp(|\Delta \mathcal{M}_k^{\tau,lin}|/t)|\mathscr{F}_{k-1}] \leq 2\} \leq C\frac{\overline{\eta}}{d}(2+M)^2 v^2 \left(1 + \frac{v}{\sqrt{d}}\right). \tag{100}$$

for all $k$. Hence once $\sqrt{d} \geq v$ we may further bound away this additional fraction incurring a further loss of a factor of 2. We may apply Lemma 5 to see that for all $t > 1$, and some absolute constant $c > 0$

$$\mathbb{P}\left(\sup_{1 \leq k \leq n} |\mathcal{M}_k^{\tau,lin} - \mathbb{E}[\mathcal{M}_0^{\tau,lin}]| > \overline{\eta}(2+M)^2 v^2 (n/d)t\right) \leq 2\exp\left(-cn\min\{t^2, t\}\right). \tag{101}$$

In the case that $n \leq dT$, this implies that there is an absolute constant so that for any $1 \leq u \leq d$

$$\sup_{1 \leq k \leq n} |\mathcal{M}_k^{\tau,lin}| \leq C\overline{\eta}(2+M)^2 v^2 \frac{\sqrt{T}}{\sqrt{d}} u \tag{102}$$

with probability at least $1 - \exp(-u)$.

## B.5 MARTINGALE FOR THE QUADRATIC TERMS

We write

$$\Delta \mathcal{M}_{k+1}^{\tau,quad} = \frac{\tilde{\eta}_k}{2d^2} \left\langle \nabla^2 q, \mathrm{clip}_{c\sqrt{d}}(\ell_k \mathbf{x}_{k+1})^{\otimes 2}\right\rangle - \frac{\tilde{\eta}_k}{2d^2}\left\langle \nabla^2 q, \mathbb{E}\left[\mathrm{clip}_{c\sqrt{d}}(\ell_k \mathbf{x}_{k+1})^{\otimes 2}|\mathscr{F}_k\right]\right\rangle \tag{103}$$

$$= T_1 + T_2, \tag{104}$$

where

$$T_1 = \frac{\tilde{\eta}_k}{2d^2}\left\langle \nabla^2 q, \mathbf{x}_{k+1}^{\otimes 2}\right\rangle \ell_k^2 \mathbb{1}_{\ell_k^2 \|\mathbf{x}_{k+1}\|^2 \leq c^2 d} - \mathbb{E}\left[\frac{\tilde{\eta}_k}{2d^2}\left\langle \nabla^2 q, \mathbf{x}_{k+1}^{\otimes 2}\right\rangle \ell_k^2 \mathbb{1}_{\ell_k^2 \|\mathbf{x}_{k+1}\|^2 \leq c^2 d}|\mathscr{F}_k\right]. \tag{105}$$

Notice that

$$\left|\left\langle \nabla^2 q, \mathbf{x}_{k+1}^{\otimes 2}\right\rangle \ell_k^2 \mathbb{1}_{\ell_k^2 \|\mathbf{x}_{k+1}\|^2 \leq c^2 d}\right| \leq \|\nabla^2 q\| c^2 d, \tag{106}$$

so that

$$|T_1| \leq c^2 \overline{\eta}^2 d^{-1}. \tag{107}$$

As for $T_2$,

$$T_2 = \frac{\tilde{\eta}_k c^2}{2d}\left\langle \nabla^2 q, \left(\frac{\mathbf{x}_{k+1}}{\|\mathbf{x}_{k+1}\|}\right)^{\otimes 2}\right\rangle \mathbb{1}_{\ell_k^2 \|\mathbf{x}_{k+1}\|^2 \geq c^2 d}$$

$$- \mathbb{E}\left[\frac{\tilde{\eta}_k c^2}{2d}\left\langle \nabla^2 q, \left(\frac{\mathbf{x}_{k+1}}{\|\mathbf{x}_{k+1}\|}\right)^{\otimes 2}\right\rangle \mathbb{1}_{\ell_k^2 \|\mathbf{x}_{k+1}\|^2 \geq c^2 d}\bigg|\mathscr{F}_k\right]. \tag{108}$$

Similarly,

$$\left|\frac{\tilde{\eta}_k c^2}{2d}\left\langle \nabla^2 q, \left(\frac{\mathbf{x}_{k+1}}{\|\mathbf{x}_{k+1}\|}\right)^{\otimes 2}\right\rangle \mathbb{1}_{\ell_k^2 \|\mathbf{x}_{k+1}\|^2 \geq c^2 d}\right| \leq \frac{\overline{\eta} c^2}{2d}. \tag{109}$$

So that overall,

$$|\Delta \mathcal{M}_{k+1}^{\tau,quad}| \leq \frac{2\overline{\eta} c^2}{d} \tag{110}$$

for all $k$. Then, by Lemma 5 we have for $t \geq 1$

$$\mathbb{P}\left(\sup_{1 \leq k \leq n} |M_k^{\tau,quad} - \mathbb{E}[M_0^{\tau,quad}]| > 2\overline{\eta}c^2(n/d)t\right) \leq 2\exp\left(-cn\min\left\{t, t^2\right\}\right). \tag{111}$$

Hence we conclude that for some absolute constant $C$ and all $1 \leq u \leq d$

$$\sup_{1 \leq k \leq n} |M_k^{\tau,quad}| \leq C\overline{\eta}c^2 \frac{\sqrt{T}}{\sqrt{d}} u \tag{112}$$

with probability at least $1 - \exp(-u)$.

## B.6 Martingale for the SDE

Recall equation (57)

$$\mathcal{M}_t^{\tau,SDE} = \int_0^t \frac{\eta(s)}{\sqrt{d}} \sqrt{2\nu(V_s^\tau)\mathcal{P}(V_s^\tau)} \nabla g(V_s^\tau)^T \sqrt{\mathbf{K}} dB_s. \tag{113}$$

We may compute the quadratic variation of $\mathcal{M}_t^{\tau,SDE}$ as

$$\langle \mathcal{M}^{\tau,SDE} \rangle_t = \int_0^t 2 \frac{\eta^2(s)}{d} \nu(V_s^\tau)\mathcal{P}(V_s^\tau)\nabla g(V_s^\tau)^T \mathbf{K} \nabla g(V_s^\tau) ds \tag{114}$$

using (89) we see that

$$\langle \mathcal{M}^{\tau,SDE} \rangle_t \leq C\overline{\eta}^2 d^{-1}(1+M)^4 t \tag{115}$$

so that

$$\sup_{0 \leq t \leq T} \langle \mathcal{M}^{\tau,SDE} \rangle_t \leq C\overline{\eta}^2 d^{-1}(1+M)^4 T \tag{116}$$

then using the sub-Gaussian tail bound for continuous martingales with bounded quadratic variation gives for $u \geq 1$

$$\mathbb{P}\left(\sup_{0 \leq s \leq T} \left|\mathcal{M}_t^{\tau,SDE}\right| > C\overline{\eta}(1+M)^2\sqrt{T}u/\sqrt{d}\right) \leq 2\exp\left(-u^2\right) \tag{117}$$

so that increasing the absolute constant $C > 0$ as needed, for all $u \geq 1$

$$\sup_{0 \leq t \leq T} \left|\mathcal{M}_t^{\tau,SDE}\right| \leq C\overline{\eta}(1+M)^2 \frac{\sqrt{T}}{\sqrt{d}} u \tag{118}$$

with probability at least $1 - \exp(-u)$.

## B.7 Bounding the error terms

The remaining technical difficulty is in bounding the error terms. We will first focus on the linear error term.

### B.7.1 Linear error terms

$$\Delta E_k^{\tau,lin} = -\frac{\tilde{\eta}_k}{d} \nabla q(v_k^\tau)^T \mathbb{E}[\text{clip}_{c\sqrt{d}}(\mathbf{x}_{k+1}\ell_k) - \mathbf{x}_{k+1}\text{clip}_c(\ell_k)] \tag{119}$$

$$= -\frac{\tilde{\eta}_k}{d} \nabla q(v_k^\tau)^T \left(\mathbb{E}\left[\mathbf{x}_{k+1}\ell_k \mathbb{1}_{\|\ell_k\mathbf{x}_{k+1}\|^2 \leq c^2 d} - \mathbf{x}_{k+1}\ell_k \mathbb{1}_{\ell_k \text{Tr}(\mathbf{K}) \leq c^2 d}\right]\right) \tag{120}$$

$$-\frac{\tilde{\eta}_k c}{\sqrt{d}} \nabla q(v_k^\tau)^T \mathbb{E}\left[\frac{\mathbf{x}_{k+1}\text{sgn}(\ell_k)}{\|\mathbf{x}_{k+1}\|} \mathbb{1}_{\|\ell_k\mathbf{x}_{k+1}\|^2 > c^2 d} - \frac{\mathbf{x}_{k+1}\text{sgn}(\ell_k)}{\sqrt{\text{Tr}(\mathbf{K})}} \mathbb{1}_{\ell_k^2 \text{Tr}(\mathbf{K}) > c^2 d}\right] \tag{121}$$

$$=: -\frac{\tilde{\eta}_k}{d} \mathbb{E}[D_k]. \tag{122}$$

For clarity, we will write $\nabla q(v_k^\tau)$ as $\nabla q_k$. We see that

$$
D_k = \begin{cases}
0, & \ell_k^2 \|\mathbf{x}_{k+1}\|^2 \le c^2 d \text{ and } \ell_k^2 \operatorname{Tr}(\mathbf{K}) \le c^2 d, \\
\nabla q_k^T \mathbf{x}_{k+1} \ell_k - \dfrac{c\sqrt{d} \nabla q_k^T \mathbf{x}_{k+1} \ell_k}{|\ell_k| \sqrt{\operatorname{Tr}(\mathbf{K})}}, & \ell_k^2 \|\mathbf{x}_{k+1}\|^2 \le c^2 d \text{ and } \ell_k^2 \operatorname{Tr}(\mathbf{K}) > c^2 d, \\
\dfrac{c\sqrt{d} \nabla q_k^T \mathbf{x}_{k+1} \ell_k}{|\ell_k| \|\mathbf{x}_{k+1}\|} - \nabla q_k^T \mathbf{x}_{k+1} \ell_k, & \ell_k^2 \|\mathbf{x}_{k+1}\|^2 > c^2 d \text{ and } \ell_k^2 \operatorname{Tr}(\mathbf{K}) \le c^2 d, \\
\dfrac{c\sqrt{d} \nabla q_k^T \mathbf{x}_{k+1} \ell_k}{|\ell_k| \|\mathbf{x}_{k+1}\|} - \dfrac{c\sqrt{d} \nabla q_k^T \mathbf{x}_{k+1} \ell_k}{|\ell_k| \sqrt{\operatorname{Tr}(\mathbf{K})}}, & \ell_k^2 \|\mathbf{x}_{k+1}\|^2 > c^2 d \text{ and } \ell_k^2 \operatorname{Tr}(\mathbf{K}) > c^2 d.
\end{cases}
\tag{123}
$$

Now, considering each case, we see that:

When $\ell_k^2 \|\mathbf{x}_{k+1}\|^2 \le c^2 d$ and $\ell_k^2 \operatorname{Tr}(\mathbf{K}) > c^2 d$, we have

$$
|D_k| = |\nabla q_k^T \mathbf{x}_{k+1} \ell_k| \left| 1 - \frac{c\sqrt{d}}{|\ell_k| \sqrt{\operatorname{Tr}(\mathbf{K})}} \right|
\tag{124}
$$

$$
\le |\nabla q_k^T \mathbf{x}_{k+1} \ell_k| \left| 1 - \frac{\|\mathbf{x}_{k+1}\|}{\sqrt{\operatorname{Tr}(\mathbf{K})}} \right|.
\tag{125}
$$

When $\ell_k^2 \|\mathbf{x}_{k+1}\|^2 > c^2 d$ and $\ell_k^2 \operatorname{Tr}(\mathbf{K}) \le c^2 d$, we have

$$
|D_k| = |\nabla q_k^T \mathbf{x}_{k+1} \ell_k| \left| \frac{c\sqrt{d}}{|\ell_k| \|\mathbf{x}_{k+1}\|} - 1 \right|
\tag{126}
$$

$$
\le |\nabla q_k^T \mathbf{x}_{k+1} \ell_k| \left| 1 - \frac{\|\mathbf{x}_{k+1}\|}{\sqrt{\operatorname{Tr}(\mathbf{K})}} \right|.
\tag{127}
$$

When $\ell_k^2 \|\mathbf{x}_{k+1}\|^2 > c^2 d$ and $\ell_k^2 \operatorname{Tr}(\mathbf{K}) > c^2 d$,

$$
|D_k| = |\nabla q_k^T \mathbf{x}_{k+1} \ell_k| \left| c\sqrt{d} \frac{1}{|\ell_k|} \right| \left| \frac{1}{\|\mathbf{x}_{k+1}\|} - \frac{1}{\sqrt{\operatorname{Tr}(\mathbf{K})}} \right|
\tag{128}
$$

$$
\le |\nabla q_k^T \mathbf{x}_{k+1} \ell_k| \left| 1 - \frac{\|\mathbf{x}_{k+1}\|}{\sqrt{\operatorname{Tr}(\mathbf{K})}} \right|.
\tag{129}
$$

Now using the numerical inequality $|1 - z| > t \implies |1 - z^2| > \max\{t, t^2\}$,

$$
\mathbb{P}\left( \left| 1 - \frac{\|\mathbf{x}_{k+1}\|}{\sqrt{\operatorname{Tr}(\mathbf{K})}} \right| > t \right) \le \mathbb{P}\left( \left| 1 - \frac{\|\mathbf{x}_{k+1}\|^2}{\operatorname{Tr}(\mathbf{K})} \right| > \max\{t, t^2\} \right).
\tag{130}
$$

Since, by assumption $\operatorname{Tr}(\mathbf{K}) = d$, and using Assumption 4 we see that, setting $s = \max\{t, t^2\}$

$$
\mathbb{P}\left( \left| 1 - \frac{\|\mathbf{x}_{k+1}\|}{\sqrt{\operatorname{Tr}(\mathbf{K})}} \right| > t \right) \le \mathbb{P}\left( |\operatorname{Tr}(\mathbf{K}) - \|\mathbf{x}_{k+1}\|^2| > ds \right)
\tag{131}
$$

$$
\le 2 \exp\left( -\min\left\{ \frac{d^2 s^2}{v^4 \|\mathbf{K}\|_F^2}, \frac{ds}{v^2 \|\mathbf{K}\|} \right\} \right).
\tag{132}
$$

Since $d^2 / \|\mathbf{K}\|_F^2 \ge d$, we conclude that for all $u \ge 1$

$$
\mathbb{P}\left( \left| 1 - \frac{\|\mathbf{x}_{k+1}\|}{\sqrt{\operatorname{Tr}(\mathbf{K})}} \right| > vu \right) \le 2 \exp\left( -du^2 \right).
\tag{133}
$$

The term $|\nabla q_k^T \mathbf{x}_{k+1} \ell_k|$ has a second moment bounded by (compare with (96))

$$\mathbb{E}|\nabla q_k^T \mathbf{x}_{k+1} \ell_k| \leq C(2+M)^2 \mathfrak{v}^2$$

for an absolute constant $C > 0$, and hence we conclude for an absolute constant $C > 0$

$$|\Delta E_k^{\tau,lin}| \leq C(2+M)^2 \overline{\eta} d^{-3/2} \mathfrak{v}^3. \tag{134}$$

Thus, taking $n \leq dT$ steps, we get

$$\max_{0 \leq k \leq n} |E_k^{\tau,lin}| \leq CT \overline{\eta} \mathfrak{v}^3 (2+M)^2 d^{-1/2}. \tag{135}$$

### B.7.2 Quadratic error terms

This follows a similar path as the linear terms. We again express

$$\Delta E_k^{\tau,quad} = \frac{\tilde{\eta}_k}{2d^2} \left( \mathbb{E}\left[ \langle \nabla^2 q, \mathbf{x}_{k+1}^{\otimes 2} \rangle \ell_k^2 \mathbb{1}_{\|\ell_k \mathbf{x}_{k+1}\|^2 \leq c^2 d} \right] - \langle \nabla^2 q, \mathbf{K} \rangle \mathbb{E}\left[ \ell_k^2 \mathbb{1}_{\ell_k^2 \operatorname{Tr}(\mathbf{K}) \leq c^2 d} \right] \right) \tag{136}$$

$$+ \frac{\tilde{\eta}_k c^2}{2d} \left( \mathbb{E}\left[ \frac{\langle \nabla^2 q, \mathbf{x}_{k+1}^{\otimes 2} \rangle}{\|\mathbf{x}_{k+1}\|^2} \mathbb{1}_{\|\ell_k \mathbf{x}_{k+1}\|^2 > c^2 d} \right] - \frac{\langle \nabla^2 q, \mathbf{K} \rangle}{\operatorname{Tr} \mathbf{K}} \mathbb{P}\left( \ell_k^2 \operatorname{Tr}(\mathbf{K}) > c^2 d \right) \right) \tag{137}$$

$$:= \frac{\tilde{\eta}_k}{2d^2} \mathbb{E}[D_k'] \tag{138}$$

with

$$D_k' = \langle \nabla^2 q, \mathbf{x}_{k+1}^{\otimes 2} \rangle \ell_k^2 \mathbb{1}_{\ell_k^2 \|\mathbf{x}_{k+1}\|^2 \leq c^2 d} - \langle \nabla^2 q, \mathbf{K} \rangle \ell_k^2 \mathbb{1}_{\ell_k^2 \operatorname{Tr}(\mathbf{K}) \leq c^2 d} \tag{139}$$

$$+ c^2 d \frac{\langle \nabla^2 q, \mathbf{x}_{k+1}^{\otimes 2} \rangle}{\|\mathbf{x}_{k+1}\|^2} \mathbb{1}_{\ell_k^2 \|\mathbf{x}_{k+1}\|^2 > c^2 d} - c^2 d \frac{\langle \nabla^2 q, \mathbf{K} \rangle}{\operatorname{Tr}(\mathbf{K})} \mathbb{1}_{\ell_k^2 \operatorname{Tr}(\mathbf{K}) > c^2 d} \tag{140}$$

$$= \begin{cases} \ell_k^2 \left( \langle \nabla^2 q, \mathbf{x}_{k+1}^{\otimes 2} \rangle - \langle \nabla^2 q, \mathbf{K} \rangle \right), & \ell_k^2 \|\mathbf{x}_{k+1}\|^2 \leq c^2 d \text{ and } \ell_k^2 \operatorname{Tr}(\mathbf{K}) \leq c^2 d, \\[2mm] \langle \nabla^2 q, \mathbf{x}_{k+1}^{\otimes 2} \rangle \ell_k^2 - c^2 d \frac{\langle \nabla^2 q, \mathbf{K} \rangle}{\operatorname{Tr}(\mathbf{K})}, & \ell_k^2 \|\mathbf{x}_{k+1}\|^2 \leq c^2 d \text{ and } \ell_k^2 \operatorname{Tr}(\mathbf{K}) > c^2 d, \\[2mm] c^2 d \frac{\langle \nabla^2 q, \mathbf{x}_{k+1}^{\otimes 2} \rangle}{\|\mathbf{x}_{k+1}\|^2} - \langle \nabla^2 q, \mathbf{K} \rangle \ell_k^2, & \ell_k^2 \|\mathbf{x}_{k+1}\|^2 > c^2 d \text{ and } \ell_k^2 \operatorname{Tr}(\mathbf{K}) \leq c^2 d, \\[2mm] c^2 d \left( \frac{\langle \nabla^2 q, \mathbf{x}_{k+1}^{\otimes 2} \rangle}{\|\mathbf{x}_{k+1}\|^2} - \frac{\langle \nabla^2 q, \mathbf{K} \rangle}{\operatorname{Tr}(\mathbf{K})} \right), & \ell_k^2 \|\mathbf{x}_{k+1}\|^2 > c^2 d \text{ and } \ell_k^2 \operatorname{Tr}(\mathbf{K}) > c^2 d. \end{cases} \tag{141}$$

Consider the function by cases. On $\ell_k^2 \|\mathbf{x}_{k+1}\|^2 \leq c^2 d$ and $\ell_k^2 \operatorname{Tr}(\mathbf{K}) > c^2 d$ we have

$$\left| \langle \nabla^2 q, \mathbf{x}_{k+1}^{\otimes 2} \rangle \ell_k^2 - c^2 d \frac{\langle \nabla^2 q, \mathbf{K} \rangle}{\operatorname{Tr}(\mathbf{K})} \right| \tag{142}$$

$$= \left| \ell_k^2 \left( \langle \nabla^2 q, \mathbf{x}_{k+1}^{\otimes 2} \rangle - \langle \nabla^2 q, \mathbf{K} \rangle \right) + \langle \nabla^2 q, \mathbf{K} \rangle \left( \ell_k^2 - \frac{c^2 d}{\operatorname{Tr}(\mathbf{K})} \right) \right| \tag{143}$$

$$\leq \ell_k^2 \left| \langle \nabla^2 q, \mathbf{x}_{k+1}^{\otimes 2} \rangle - \langle \nabla^2 q, \mathbf{K} \rangle \right| + \langle \nabla^2 q, \mathbf{K} \rangle \frac{\ell_k^4}{c^2 d} \left| \operatorname{Tr}(\mathbf{K}) - \|\mathbf{x}_{k+1}\|^2 \right|. \tag{144}$$

Similarly, if $\ell_k^2 \|\mathbf{x}_{k+1}\|^2 > c^2 d$ and $\ell_k^2 \operatorname{Tr}(\mathbf{K}) \leq c^2 d$

$$\left| c^2 d \frac{\langle \nabla^2 q, \mathbf{x}_{k+1}^{\otimes 2} \rangle}{\|\mathbf{x}_{k+1}\|^2} - \langle \nabla^2 q, \mathbf{K} \rangle \ell_k^2 \right| \tag{145}$$

$$= \left| \frac{c^2 d}{\|\mathbf{x}_{k+1}\|^2} \left( \langle \nabla^2 q, \mathbf{x}_{k+1}^{\otimes 2} \rangle - \langle \nabla^2 q, \mathbf{K} \rangle \right) + \langle \nabla^2 q, \mathbf{K} \rangle \left( \frac{c^2 d}{\|\mathbf{x}_{k+1}\|^2} - w^2 \right) \right| \tag{146}$$

$$\leq \ell_k^2 \left| \langle \nabla^2 q, \mathbf{x}_{k+1}^{\otimes 2} \rangle - \langle \nabla^2 q, \mathbf{K} \rangle \right| + \langle \nabla^2 q, \mathbf{K} \rangle \frac{\ell_k^4}{c^2 d} \left| \operatorname{Tr}(\mathbf{K}) - \|\mathbf{x}_{k+1}\|^2 \right|. \tag{147}$$

and finally when $\ell_k^2 \|\mathbf{x}_{k+1}\|^2 > c^2 d$ and $\ell_k^2 \operatorname{Tr}(\mathbf{K}) > c^2 d$ we have

$$\left| c^2 d \left( \frac{\langle \nabla^2 q, \mathbf{x}_{k+1}^{\otimes 2} \rangle}{\|\mathbf{x}_{k+1}\|^2} - \frac{\langle \nabla^2 q, \mathbf{K} \rangle}{\operatorname{Tr}(\mathbf{K})} \right) \right| \le \ell_k^2 \left| \langle \nabla^2 q, \mathbf{x}_{k+1}^{\otimes 2} \rangle - \langle \nabla^2 q, \mathbf{K} \rangle \right| \tag{148}$$

$$+ \langle \nabla^2 q, \mathbf{K} \rangle \frac{\ell_k^4}{c^2 d} \left| \operatorname{Tr}(\mathbf{K}) - \|\mathbf{x}_{k+1}\|^2 \right| \mathbb{1}_{\ell_k^2 \operatorname{Tr}(\mathbf{K}) > c^2 d}. \tag{149}$$

So overall,

$$|D_k'| \le \ell_k^2 \left| \langle \nabla^2 q, \mathbf{x}_{k+1}^{\otimes 2} \rangle - \langle \nabla^2 q, \mathbf{K} \rangle \right| + \langle \nabla^2 q, \mathbf{K} \rangle \frac{\ell_k^4}{c^2 d} \left| \operatorname{Tr}(\mathbf{K}) - \|\mathbf{x}_{k+1}\|^2 \right| \mathbb{1}_{\ell_k^2 \operatorname{Tr}(\mathbf{K}) > c^2 d}. \tag{150}$$

Now, we may use the Hanson-Wright inequality (Assumption 4) along with the inequality

$$\|\sqrt{\mathbf{K}} \nabla^2 q \sqrt{\mathbf{K}}\|_F^2 \le \operatorname{Tr}(\mathbf{K}) \|\nabla^2 q\|^2 \le \operatorname{Tr}(\mathbf{K}) = d \tag{151}$$

to see that for all $t \ge 0$

$$\mathbb{P} \left( \left| \langle \nabla^2 q, \mathbf{x}_{k+1}^{\otimes 2} \rangle - \langle \nabla^2 q, \mathbf{K} \rangle \right| > t \varrho^2 \right) \le 2 \exp \left( - \min \left\{ \frac{t^2}{d}, t \right\} \right). \tag{152}$$

We also recall from (132) that

$$\mathbb{P} \left( \left| \|\mathbf{x}_{k+1}\|^2 - \operatorname{Tr}(\mathbf{K}) \right| > t \varrho^2 \right) \le 2 \exp \left( - \min \left\{ \frac{t^2}{d}, t \right\} \right).$$

Hence overall, we conclude that for some absolute constant $C > 0$

$$|\mathbb{E} D_k'| \le C \varrho^4 (1 + M)^2 \sqrt{d}.$$

So that overall

$$\left| \Delta E_k^{\tau, quad} \right| \le C \overline{\eta} \varrho^4 (1 + M)^2 d^{-3/2}. \tag{153}$$

and summing over $k \le n \le Td$,

$$\max_{0 \le k \le n} \left| E_k^{\tau, quad} \right| \le C T \overline{\eta} \varrho^4 (1 + M)^2 d^{-1/2}. \tag{154}$$

This completes the proof of Lemma 2.

## C  ADDITIONAL EXPERIMENTS AND SUPPLEMENTARY FIGURES

### C.1  STUDENT-T DISTRIBUTED NOISE

Here, we examine the influence of heavy-tailed noise on the (CSC) and (CCC) using the Student-t family of distributions.

The Student-t distribution, characterized by its degrees of freedom (df) parameter, allows us to explore a continuum of tail behaviors. As the degrees of freedom decrease, the distribution becomes more heavy-tailed, ranging from the Cauchy distribution (df = 1) to the Gaussian distribution as df approaches infinity.

To better understand how varying tail behaviors affect the dynamics of clipped SGD, we generate plots of these thresholds as the degrees of freedom parameter changes. These results are in Figure 5.

As df increases, so to do values of the (CSC) and (CCC) ratios. This is unsurprising as the distribution becomes more heavy-tailed.

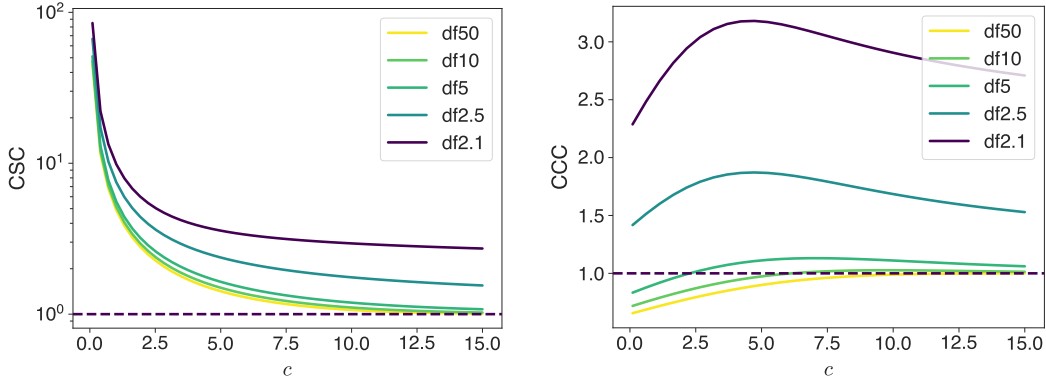

Figure 5: The `CSC` and `CCC` where the noise is Student-t distributed. We hold the variance fixed and vary over the degrees-of-freedom (DOF) parameter. Notice how for high DOF, the thresolds resemble Gaussian behaviour (compare to Figure 2 in the main paper). Meanwhile for small DOF, we see that both the `CSC` and `CCC` are high, suggesting clipping is particularly effective. This reflects the heavy-tailed behaviour of the small DOF Student-t distribution.

## C.2 MEASURING $\mu$ AND $\nu$ IN REAL NETWORKS

We can estimate $\mu$ and $\nu$ in the non-linear setting as well. Even though C-HSGD does not fully capture the dynamics in this regime, computing estimates of $\mu$ and $\nu$ may give useful information about how to apply clipping schedules. Given a loss function $\mathcal{L}(\boldsymbol{\theta})$ induced by some non-linear model $\mathbf{z}(\boldsymbol{\theta})$, the generalization of Equation 7 is given by

$$\mu_c(\boldsymbol{\theta}) = \frac{\|\mathbb{E}[\mathrm{clip}_c(\nabla_{\boldsymbol{\theta}}\mathcal{L})]\|}{\|\mathbb{E}[\nabla_{\boldsymbol{\theta}}\mathcal{L}]\|} \qquad \text{and} \qquad \nu_c(\boldsymbol{\theta}) = \frac{\mathbb{E}[\|\mathrm{clip}_c(\nabla_{\boldsymbol{\theta}}\mathcal{L})\|^2]}{\mathbb{E}[\|\nabla_{\boldsymbol{\theta}}\mathcal{L}\|^2]}. \qquad (155)$$

Here the averages are across sampled minibatches. These forms can be derived from the original calculation by making the substitution $\mathbf{x} \rightarrow \mathbf{J}$, where $\mathbf{J}$ is the Jacobian $\frac{\partial \mathbf{z}}{\partial \boldsymbol{\theta}}$.

The numerator and denominator can be estimated online using a running average (e.g. exponential moving average) updated after every C-SGD step. This does not require any additional backpropagation steps, but does require keeping 2 extra running averages in the shape of $\boldsymbol{\theta}$ to compute $\mu_c$. However, if we use the form of $\mu_c$ from Equation 9, adapted as

$$\mu_c(\boldsymbol{\theta}) = \mathbb{P}[\|\nabla_{\boldsymbol{\theta}}\mathcal{L}\| \leq c] \qquad (156)$$

(that is, the probability of unclipped gradients), no additional memory costs are incurred and the computation is efficient.

We used this estimator to compute $\mu$ and $\nu$ for ResNet18 and ViT S/16 trained on CIFAR10 (Figure 6, left column). The models were trained without clipping, and $\mu$ and $\nu$ were computed using different clipping values $c$. With a larger clipping threshold of $c = 10$, gradients are rarely clipped and $\mu(t) \approx \nu(t) \approx 1$; there is non-trivial dynamics for smaller clipping strength. We also computed the (`CSC`) and (`CCC`) for the different thresholds (Figure 6, right column). For ResNet18, the (`CCC`) is above 1 for the smaller thresholds; if the theory holds, then clipping is beneficial here. In contrast, for ViT it remains at or below 1 and clipping would not help.

Further investigation is needed to understand the utility of using $\mu$ and $\nu$ in the non-linear setting; the two main obstacles are the fidelity of the H-CSGD approximation and, relatedly, the question of whether or not the max-(`CCC`) is strictly better than the unclipped schedule in the non-linear setting. Non-convex optimization can often have more effects which depend on the whole training trajectory; the form of the H-CSGD dynamics in the linear setting allowed us to largely ignore these with the appropriate choice of $\mu$ and $\nu$. We leave the exploration and development of practical applications of $\mu$ and $\nu$ to set joint learning rate and clipping schedules to future work.

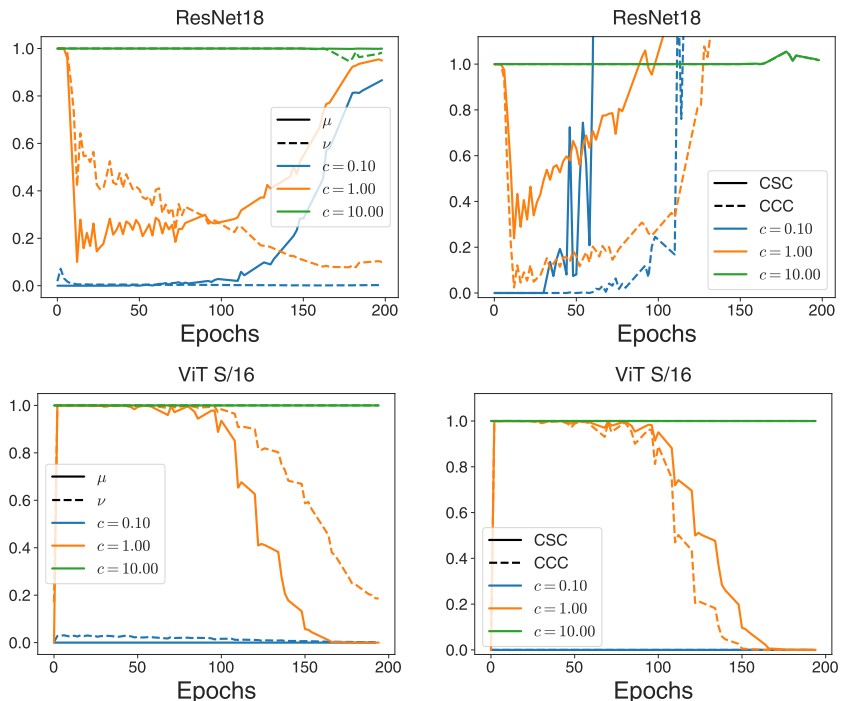

Figure 6: $\mu$, $\nu$, CSC and CCC for the realistic models of Appendix C (ResNet18 and ViT) trained on CIFAR10. $\mu$ and $\nu$ were computed using Equations 155 and the additional approximation from Equation 156. $\mu$ and $\nu$ are computed for three log-spaced clipping thresholds. Our theory suggests that ResNet may benefit from clipping, but ViT in this setting does not.

### C.3 MEASURING THE INTRINSIC DIMENSION IN REAL NETWORKS

Recall that the intrinsic dimension $d$ is defined in terms of the spectrum of $\mathbf{K}$:

$$d := \mathrm{Tr}(\mathbf{K})/\|\mathbf{K}\|.$$

We can extend the definition of the intrinsic dimension to the non-linear setting by considering a linearization of the dynamics. Given a loss function $\mathcal{L}(\mathbf{z})$ and a model $\mathbf{z}(\boldsymbol{\theta})$ on parameters $\boldsymbol{\theta}$, the Gauss Newton matrix $\mathbf{G}$ of the loss is defined by:

$$\mathbf{G} := \nabla_{\boldsymbol{\theta}}\mathbf{z}^{\top}\nabla_{\mathbf{z}}^2\mathcal{L}\nabla_{\boldsymbol{\theta}}\mathbf{z}. \tag{157}$$

Here $\nabla_{\boldsymbol{\theta}}\mathbf{z}$ is the model Jacobian, and $\nabla_{\mathbf{z}}^2\mathcal{L}$ is the Hessian of the loss with respect to the model outputs. $\mathbf{G}$ encodes the second derivative of the loss with respect to a linearized model $\tilde{\mathbf{z}} = \mathbf{z}(\boldsymbol{\theta}_0) + \nabla_{\boldsymbol{\theta}}\mathbf{z}(\boldsymbol{\theta} - \boldsymbol{\theta}_0)$.

For a linear model on MSE loss (as we studied in the main text), we have $\mathbf{K} = \mathbf{G}$. If we took a non-linear model during training, and locally linearized the model and loss, we would measure the intrinsic dimension with $\mathbf{G}$ as well. Therefore, on non-linear models, we will define

$$d_{nl} := \mathrm{Tr}(\mathbf{G})/\|\mathbf{G}\| \tag{158}$$

as the *non-linear intrinsic dimension*.

With this definition, we can measure the intrinsic dimension on neural network models during training. We measured $d_{nl}$ on ResNet18 He et al. (2016) and ViT S/16 Dosovitskiy et al. (2020) for networks trained on CIFAR10 using MSE loss (Figure 7). We see that for ResNet18, $d_{nl}$ increases from $\sim 100$ to $10^3$, while for ViT $d_{nl}$ stays steady at $\sim 300$. In both cases $d_{nl}$ is large, but it is very model dependent.

This suggests that real neural network models are in the effectively high-dimensional regime; we leave to future work the question of which concepts from the basic theory generalize to the non-linear setting.

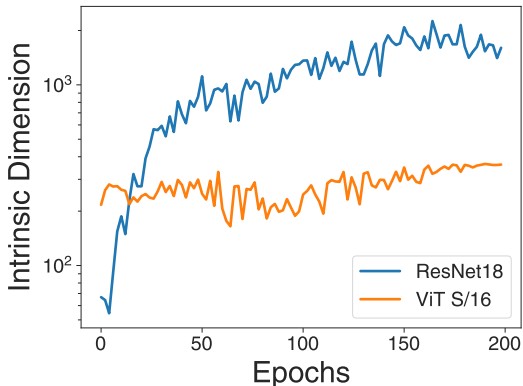

Figure 7: Non-linear intrinsic dimension $d_{nl}$ for models trained on MSE loss. For ResNet18 (blue), $d_{nl}$ increases by a factor of 10 over training, while for ViT S/16 (orange) $d_{nl}$ remains relatively constant.

### C.4 FURTHER EXPERIMENTS WITH THE HEURISTIC CLIPPING SCHEDULE

In this section we showcase some preliminary results investigating the proposed heuristic optimal clipping schedule given in Section 5.3. Note that a full investigation of the heuristic optimal clipping schedule across many datasets and in differing models is beyond the scope of this paper.

#### C.4.1 HEURISTIC CLIPPING SCHEDULE ON REAL DATA

In this section we showcase some results on applying the heuristically optimal learning rate schedule for real datasets, namely the Wikitext2 dataset for next-token prediction. For experimental details on how the dataset was embedded and key quantities were estimated see Appendix H. A full investigation into the efficacy of this heuristic schedule will require extensive experimentation which is beyond the scope of this work. Here we simply provide evidence that the schedule and heuristics are effective even in real world conditions.

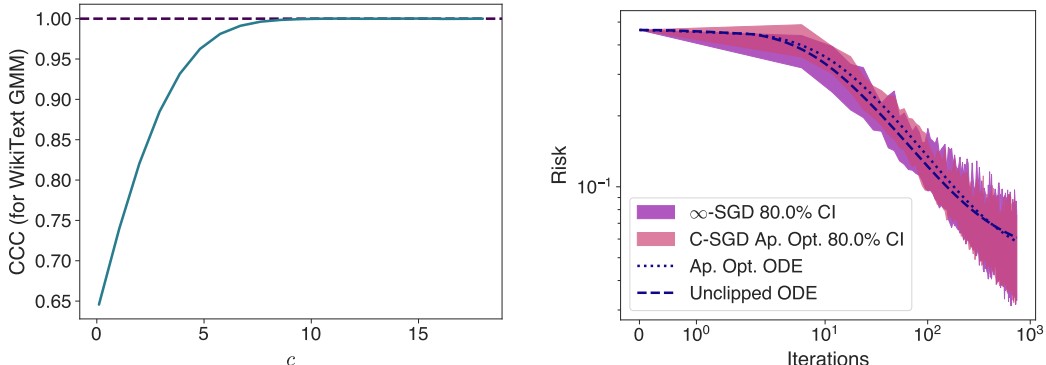

Figure 8: We train a linear model using the heuristic optimal clipping and learning rate schedules given in Section 5.3. The (CCC) is given in the left hand figure. Note that the optimal clipping in this scenario is not to clip. This is captured by the heuristic optimal. In fact, heuristic optimal actually performs slightly better due to imperfectly compensating the learning rate.

#### C.4.2 INSENSITIVITY TO THE TUNED CONSTANT

Here we investigate the sensitivity of the heuristic clipping schedule to the tuned hyper-parameter $\kappa$. In Figure 9 we first find the optimal $\kappa$ through a simple grid search. To investigate the sensitivity, we then run clipped SGD with the heuristic schedule with both $1.5\kappa$ and $\kappa/2$. Despite the relatively large change in the hyper-parameter we observe a relatively small stray from the behaviour under $\kappa$.

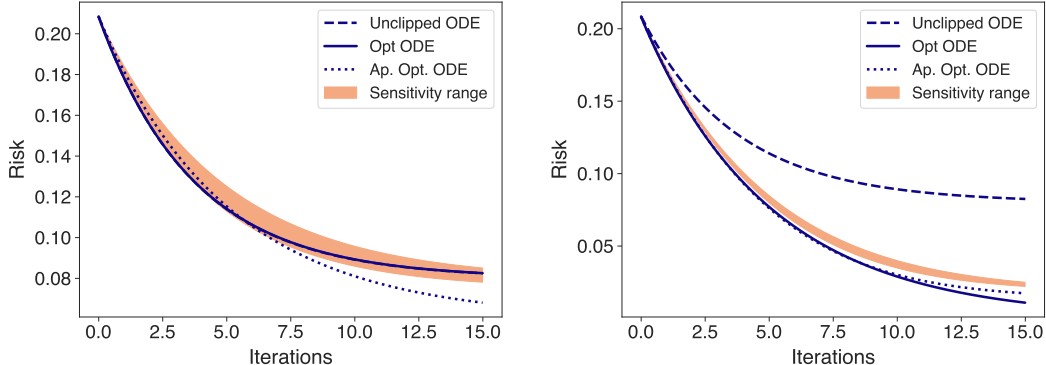

Figure 9: Sensitivity experiment of the heuristic clipping schedule (Equation (25) to changes in the optimal $\kappa$. Here the sensitivity range gives the values when run with $1.5\kappa$ and $\kappa/2$.

## D    SOME EXAMPLES OF $\mu$ AND $\nu$

In this section we give some examples of $\mu$ and $\nu$ as defined in equation (7) under various common distributions. First, we will describe how to define $\mu_c$ and $\nu_c$ as functions of the risk.

Notice that, with Gaussian data, $\ell_{\boldsymbol{\theta}} \stackrel{\text{law}}{=} \sqrt{2\mathcal{R}(\boldsymbol{\theta})}\xi - \epsilon$ where $\xi$ is a standard Gaussian. Thus we can define $\tilde{\mu}$ and $\tilde{\nu}$ such that

$$\tilde{\mu}(\boldsymbol{\Theta}_t) = \frac{\|\mathbb{E}[\text{clip}_c(\sqrt{2\mathcal{R}(\boldsymbol{\Theta}_t)}\xi - \epsilon)\mathbf{x}]\|}{\|\mathbb{E}[(\sqrt{2\mathcal{R}(\boldsymbol{\Theta}_t)}\xi - \epsilon)\mathbf{x}]\|} \qquad \tilde{\nu}(\boldsymbol{\Theta}_t) = \frac{\mathbb{E}[\text{clip}_c^2(\sqrt{2\mathcal{R}(\boldsymbol{\Theta}_t)}\xi - \epsilon)]}{\mathbb{E}[(\sqrt{2\mathcal{R}(\boldsymbol{\Theta}_t)}\xi - \epsilon)^2]} \tag{159}$$

and $\mu_c(\boldsymbol{\Theta}_t) = \tilde{\mu}_c(\mathcal{R}(\boldsymbol{\Theta}_t))$ and $\nu_c(\boldsymbol{\Theta}_t) = \tilde{\nu}_c(\mathcal{R}(\boldsymbol{\Theta}_t))$. In what follows $r = \mathcal{R}(\boldsymbol{\theta})$ for some $\boldsymbol{\Theta}_t$. In the following examples, we will simplify notation and simply let $\mathcal{R}(\boldsymbol{\Theta}_t) = r$.

### D.1    GAUSSIAN DATA AND GAUSSIAN NOISE

Consider $a \sim N(0, \mathbf{K})$ and $\epsilon \sim N(0, \sigma^2)$. First define

$$F(z) = \text{erf}\left(\frac{z}{\sqrt{2}}\right) - \sqrt{\frac{2}{\pi}} z e^{-z^2/2} \tag{160}$$

Then, we have

$$\mu_c(r) = \text{erf}\left(\frac{c}{\sqrt{4(r + \sigma^2/2)}}\right) \tag{161}$$

$$(2r + \sigma^2)\nu_c(r) = 2rF\left(\frac{c}{\sqrt{2(r + \eta^2/2)}}\right) + c^2 \text{erfc}\left(\frac{c}{\sqrt{4(r + \eta^2/2)}}\right) \tag{162}$$

### D.2    GAUSSIAN DATA AND RADEMACHER-LIKE NOISE

$$\epsilon_k = \begin{cases} -\lambda & \text{with probability } q/2 \\ 0 & \text{with probability } 1 - q \\ \lambda & \text{with probability } q/2 \end{cases} \tag{163}$$

Note that $\sigma^2 = \text{Var}(\epsilon) = \lambda^2 q$. For some standard Gaussian random variable $z$, $\mu$ and $\nu$ may be computed as

$$\mu_c(r) = q\mathbb{P}\left(|z - \lambda| \leq \frac{c}{\sqrt{2r}}\right) + (1 - q)\mathbb{P}\left(|z| \leq \frac{c}{\sqrt{2r}}\right) \tag{164}$$

$$= \frac{q}{2}\left(\mathrm{erf}\left(\frac{c - \lambda}{\sqrt{4r}}\right) + \mathrm{erf}\left(\frac{c + \lambda}{\sqrt{4r}}\right)\right) + (1 - q)\,\mathrm{erf}\left(\frac{c}{\sqrt{4r}}\right) \tag{165}$$

$$(2r + \sigma^2)\nu_c(r) = 2r\frac{q}{2}\left(F\left(\frac{c - \lambda}{\sqrt{2r}}\right) + F\left(\frac{c + \lambda}{\sqrt{2r}}\right)\right) + 2r(1 - q)F\left(\frac{c}{\sqrt{2r}}\right) \tag{166}$$

$$+ \frac{q\lambda}{\sqrt{\pi r}}\left(\exp\left(-\frac{(c + \lambda)^2}{2r}\right) - \exp\left(-\frac{(c - \lambda)^2}{2r}\right)\right) \tag{167}$$

$$+ \frac{q\lambda^2}{2}\left(\mathrm{erf}\left(\frac{c - \lambda}{\sqrt{4r}}\right) + \mathrm{erf}\left(\frac{c + \lambda}{\sqrt{4r}}\right)\right) \tag{168}$$

$$+ \frac{qc^2}{2}\left(\mathrm{erfc}\left(\frac{c - \lambda}{\sqrt{2r}}\right) + \mathrm{erfc}\left(\frac{c + \lambda}{\sqrt{2r}}\right) + (1 - q)\mathbb{P}\left(|z| > \frac{c}{\sqrt{2r}}\right)\right) \tag{169}$$

## D.3 GAUSSIAN DATA AND UNIFORM NOISE

For $a \sim N(0, \mathbf{K})$ and uniform noise supported on $[-M, M]$ we have $\sigma^2 = M^2/3$ and

$$\mu_c(r) = 1 - \frac{1}{2M}c^2\left(e^{-\frac{(c + M)^2}{4r}}(e^{\frac{cM}{r}} - 1)\sqrt{\frac{4r}{\pi}}\right) \tag{170}$$

$$- \frac{1}{2M}c^2\left((M - c)\,\mathrm{erfc}\left(\frac{c - M}{\sqrt{4r}}\right) + (c + M)\,\mathrm{erfc}\left(\frac{c + M}{\sqrt{4r}}\right)\right) \tag{171}$$

$$(2r + \sigma^2)\nu_c(r) = \frac{-1}{6M}e^{-\frac{(c + M)^2}{4r}}\sqrt{\frac{4r}{\pi}}\left(-c^2 + cM - M^2 - 4r + e^{cM/r}(c^2 + cM + M^2 + 4r)\right) \tag{172}$$

$$- \frac{1}{6M}(c^3 - M^3 - 6Mr)\,\mathrm{erf}\left(\frac{c - M}{\sqrt{4r}}\right) + \frac{1}{6M}(c^3 + M^3 + 6Mr)\,\mathrm{erf}\left(\frac{c + M}{\sqrt{4r}}\right) \tag{173}$$

$$+ \frac{1}{2M}c^2\left(e^{-\frac{(c + M)^2}{4r}}(e^{\frac{cM}{r}} - 1)\sqrt{\frac{2\sigma^2}{\pi}}\right) \tag{174}$$

$$+ \frac{1}{2M}c^2\left((M - c)\,\mathrm{erfc}\left(\frac{c - M}{\sqrt{4r}}\right) + (c + M)\,\mathrm{erfc}\left(\frac{c + M}{\sqrt{4r}}\right)\right) \tag{175}$$

## D.4 GAUSSIAN DATA AND SYMMETRIC EXPONENTIAL NOISE

For $a \sim N(0, \mathbf{K})$ and symmetric exponential noise, also known as Laplacian, with density

$$f(x) = \lambda e^{-|x|\lambda}/2 \tag{176}$$

then we have

$$\mu_c(r) = 2\,\mathrm{erf}\left(\frac{c}{\sqrt{2r^2}}\right) \tag{177}$$

$$+ e^{\lambda(-2c + \lambda r^2)/2}(e^{2c\lambda} - \mathrm{erf}\left(\frac{(c - \lambda r^2)}{\sqrt{2r^2}}\right) - e^{2c\lambda}\,\mathrm{erf}\left(\frac{(c + \lambda r^2)}{\sqrt{2r^2}}\right) - 1)/2 \tag{178}$$

Then, if $T_c(r)$ satisfies

$$(8 + 4\lambda^2 r^2)T(r) = 2e^{\lambda(2c+\lambda r^2)/2}(2 - 2c\lambda + c^2\lambda^2) - 2e^{\lambda(-2c+\lambda r^2)/2}(2 + 2c\lambda + c^2\lambda^2) \quad (179)$$

$$- 4ce^{\frac{-c^2}{2r^2}}\lambda^2\sqrt{2r^2/\pi} + 2\lambda^2 r^2 + 2\lambda^2 r^2(\mathrm{erf}\left(\frac{c}{\sqrt{2r^2}}\right) - 1) \quad (180)$$

$$+ (4 + 2\lambda^2 r^2 + 8)\,\mathrm{erf}\left(\frac{c}{\sqrt{2r^2}}\right) \quad (181)$$

$$- 2e^{\lambda(-2c+\lambda r^2)/2}(2 + 2c\lambda + c^2\lambda^2)\,\mathrm{erf}\left(\frac{c - \lambda r^2}{\sqrt{2r^2}}\right) \quad (182)$$

$$- 2e^{\lambda(2c+\lambda r^2)/2}(2 - 2c\lambda + c^2\lambda^2)\,\mathrm{erf}\left(\frac{c + \lambda r^2}{\sqrt{2r^2}}\right) \quad (183)$$

we have

$$\nu_c(r) = T_c(r) + c^2(1 - \mu_c(r))/(2r + \sigma^2) \quad (184)$$

## E SIMPLIFICATION OF $\mu$ AND $\nu$

**Theorem 8.** *Let* $\mathbf{x} \sim N(0, \mathbf{K})$ *and let the noise be* $\sigma\epsilon$ *for a fixed random variable* $\epsilon$ *and* $\sigma > 0$. *Assume that the characteristic function of* $\epsilon$, $\varphi(t) = \mathbb{E}[e^{it\epsilon}]$ *is integrable. This assumption implies that* $\epsilon$ *has a continuous density* $g(y)$. *We will additionally assume that* $g(0) > 0$.

*Then, there exist absolute constants* $\kappa_l, \kappa_u$ *depending on the distribution of the noise (but not* $\sigma$*) such that,*

$$\kappa_l \min\left(1, \frac{c}{\sqrt{2R + \sigma^2}}\right) \leq \mu_c(R) \leq \kappa_u \min\left(1, \frac{c}{\sqrt{2R + \sigma^2}}\right) \quad (185)$$

*Proof.* We first complete the upper bound. Let $g \sim N(0,1)$. Notice that $\ell_{\boldsymbol{\theta}} \stackrel{\text{law}}{=} \sqrt{2\mathcal{R}(\boldsymbol{\theta})}g - \sigma\epsilon$. Thus,

$$\mu_c(\boldsymbol{\theta}) = \mathbb{P}(|\ell_{\boldsymbol{\theta}}| \leq c). \quad (186)$$

Going forward, we omit the dependence on $\boldsymbol{\theta}$ for clarity. Let $f$ be the density of $\ell$. Then, by the Fourier inversion theorem

$$\begin{aligned}f(x) &= \frac{1}{2\pi}\int e^{-ixt}e^{-\mathcal{R}t^2}\varphi(\sigma t)\mathrm{d}t \\ &\leq \frac{1}{2\pi}\int e^{-\mathcal{R}t^2}|\varphi(\sigma t)|\mathrm{d}t,\end{aligned} \quad (187)$$

where $\varphi(t)$ is the characteristic function of $\epsilon$. Now consider cases: when $2\mathcal{R} \geq \sigma^2$,

$$\begin{aligned}f(x) &\leq \frac{1}{2\pi\sigma}\int e^{-\mathcal{R}t^2/\sigma^2}|\varphi(t)|dt \\ &\leq \pi^{-1/2}\frac{1}{\sqrt{2\mathcal{R} + \sigma^2}}\end{aligned} \quad (188)$$

Meanwhile, if $\sigma^2 < 2\mathcal{R}$

$$f(x) \leq \frac{1}{2\pi\sigma} \int e^{-\mathcal{R}t^2/\sigma^2} |\varphi(t)| dt$$

$$\leq \frac{\int |\varphi(t)| dt}{\sqrt{2\pi}} \frac{1}{\sqrt{\sigma^2 + 2\mathcal{R}}} \tag{189}$$

By assumption, $\int |\varphi(t)| dt < \infty$ and while it depends on the noise distribution it does not depend on the variance/scale parameter $\sigma$. Since,

$$\mu_c = \int_{-c}^{c} f(x) dx, \tag{190}$$

putting the two cases together completes the bound.

We now proceed with the lower bound. Let $A \geq 1$ and again consider cases. First, let $\sigma < Ac$ and $\sqrt{2R} < Ac$ for some $A$. Then,

$$\begin{aligned}
\mu_c &= \mathbb{P}(|\ell_{\boldsymbol{\theta}}| \leq c) \\
&= \mathbb{P}(|\sqrt{2R}g + \sigma\epsilon| \leq c) \\
&\geq \mathbb{P}(|g| \leq 1/2A)\mathbb{P}(|\epsilon| \leq 1/2A) \\
&= \kappa_A > 0.
\end{aligned} \tag{191}$$

Where $\kappa_A$ is bounded away from $0$ by assumption. And so,

$$\mu_c \geq \kappa_A \min\left(1, \frac{c}{\sqrt{\sigma^2 + 2R}}\right). \tag{192}$$

Now for the remaining cases. We can express $\mu_c$ by integrating over the density of $\ell_{\boldsymbol{\theta}}$. Let $\mathcal{L}$ represent the law of $\sigma\epsilon$. Then,

$$\mu_c = \frac{1}{\sqrt{4\pi R}} \int_{-c}^{c} \int_{\mathbb{R}} e^{-(x+y)^2/4R} d\mathcal{L}(y) \, dx. \tag{193}$$

Consider the case where $\sqrt{2R} > Ac$ and $\sqrt{2R} > \sigma$. Then,

$$\begin{aligned}
\mu_c &\geq \frac{1}{\sqrt{4\pi R}} \int_{-c}^{c} \int_{-1/A}^{1/A} e^{-(x+y\sigma)^2/4R} g(y) dy \, dx \\
&\geq \min\left(1, \frac{c}{\sqrt{2R + \sigma^2}}\right) \frac{4me^{-1/8A^2}}{A\sqrt{2\pi}}
\end{aligned} \tag{194}$$

where we may lower bound the integrand since $|(x+y\sigma)^2/4R| \leq 1/2A$ over the rectangle $(x, y) \in [-c, c] \times [-1/A, 1/A]$. Similarly, we consider the case $\sqrt{2R} < \sigma$ and $\sigma > Ac$. Then,

$$\begin{aligned}
\mu_c &\geq \frac{1}{\sigma\sqrt{2\pi}} \int_{-c}^{c} \int_{-1/A}^{1/A} e^{-y^2/2} g\left(\frac{\sqrt{2R}y - x}{\sigma}\right) dy \, dx \\
&\geq \min\left(1, \frac{c}{\sqrt{2R + \sigma^2}}\right) \frac{4me^{-1/2A^2}}{A\sqrt{2\pi}}.
\end{aligned} \tag{195}$$

Choosing $\kappa_l = \min\left(\kappa_A, \frac{4me^{-1/8A^2}}{A\sqrt{2\pi}}\right)$ completes the proof. $\square$

**Corollary 1.** *In the same setting as Theorem 8,*

$$\kappa_l \min\left(1, \frac{c^2}{2R + \sigma^2}\right) \leq \nu_c^2(R) \leq \kappa_u \min\left(1, \frac{c^2}{2R + \sigma^2}\right). \tag{196}$$

*Proof.* $\nu_c$ is trivially less than or equal to 1. Recall the definition,

$$
\begin{aligned}
\nu_c(R) &= \frac{1}{2R + \sigma^2} \left( \mathbb{E}\left[\ell_{\boldsymbol{\theta}}^2 \mathbb{1}_{|\ell_{\boldsymbol{\theta}}| \leq c}\right] + c^2(1 - \mu_c) \right) \\
&\leq \frac{c^2}{2R + \sigma^2}
\end{aligned}
\tag{197}
$$

which gives the upper bound with coefficient 1. Now for the lower bound. We can write,

$$
\begin{aligned}
\nu_c &= \frac{1}{2R + \sigma^2} \left( \int_0^\infty \mathbb{P}(\ell_{\boldsymbol{\theta}}^2 \mathbb{1}_{|\ell_{\boldsymbol{\theta}}| \leq c} \geq t)\,\mathrm{d}t + c^2(1 - \mu_c) \right) \\
&= \frac{1}{2R + \sigma^2} \left( \int_0^{c^2} \mathbb{P}(\ell_{\boldsymbol{\theta}}^2 \geq t)\,\mathrm{d}t + c^2(1 - \mu_c) \right) \\
&\geq \frac{c^2}{2R + \sigma^2} 2(1 - \mu_c)
\end{aligned}
\tag{198}
$$

So we can now use the $\mu_c$ bound derived above to obtain the desired bound for $\nu_c$.

$\square$

## F   PROOF OF STABILITY AND EFFECTIVENESS THEOREMS

### F.1   PROOF OF THEOREM 2

To see there exists $c > 0$ such that (CSC) holds, it is simpler to work with the inverse of the ratio. We remark that

$$
\lim_{c \to 0^+} \frac{\nu_c}{\mu_c} = \lim_{c \to 0^+} \frac{\mathbb{E}(\ell^2 \mathbb{1}_{|\ell| \leq c} + c^2 \mathbb{1}_{|\ell| > c})}{\mathbb{P}(|\ell| \leq c)} \tag{199}
$$

$$
= \lim_{c \to 0^+} \frac{1}{\mathbb{P}(|\ell| \leq c)} \int_{|\ell| \leq c} \ell^2\,d\mathbb{P} + \frac{c^2(1 - \mathbb{P}(|\ell| \leq c))}{\mathbb{P}(|\ell| \leq c)} \tag{200}
$$

$$
= \lim_{c \to 0^+} \frac{1}{\mathbb{P}(|\ell| \leq c)} \int_{|\ell| \leq c} \ell^2\,d\mathbb{P} + \frac{c^2}{\mathbb{P}(|\ell| < c)}. \tag{201}
$$

Therefore, it suffices to show that (201) is less than 1. Indeed, the former term converges to 0 by the Lebesgue-Differentiation Theorem. For the latter, let us assume that $\epsilon \sim \pi$ for some probability-measure $\pi$. Given that $\mathbf{x} \sim N(0, \mathbf{K})$, let us denote $f$ to be the (Gaussian) density of $\langle \mathbf{x}, \boldsymbol{\theta} - \boldsymbol{\theta}^* \rangle$. It follows that

$$
\mathbb{P}(|\ell| \leq c) = \mathbb{P}(-c + \epsilon \leq \langle \mathbf{x}, \boldsymbol{\theta} - \boldsymbol{\theta}^* \rangle \leq c + \epsilon) \tag{202}
$$

$$
= \int_{\mathbb{R}} \int_{-c+\epsilon}^{c+\epsilon} f(x)\,dx\,d\pi(\epsilon). \tag{203}
$$

Differentiating with respect to $c$ yields,

$$
\frac{d}{dc} \left( \int_{\mathbb{R}} \int_{-c+\epsilon}^{c+\epsilon} f(x)\,dx\,d\pi(\epsilon) \right) = \int_{\mathbb{R}} f(c + \epsilon) + f(-c + \epsilon)\,d\pi(\epsilon). \tag{204}
$$

By L'Hôpital's rule, the latter term of (201) becomes

$$
\lim_{c \to 0} \frac{c^2}{\mathbb{P}(|\ell| < c)} = \lim_{c \to 0} \frac{2c}{\int_{\mathbb{R}} f(c + \epsilon) + f(-c + \epsilon)\,d\pi(\epsilon)} \tag{205}
$$

$$
= \lim_{c \to 0^+} \frac{2c}{\int_{\mathbb{R}} 2f(\epsilon)\,d\pi(\epsilon)} \tag{206}
$$

$$
= 0. \tag{207}
$$

### F.2 PROOF OF THEOREM 3

First, notice that $\sigma^2 = q\lambda^2$. In the limit as $|R_t| \to 0$ we have,

$$\mu(R_t) = \mathbb{P}(|\epsilon| \leq c) \tag{208}$$

$$= \begin{cases} 1 & \lambda \leq c \\ 1 - q & \lambda > c \end{cases} \tag{209}$$

$$\nu(R_t) = \begin{cases} 1 & \lambda \leq c \\ c^2/\lambda^2 & \lambda > c \end{cases} \tag{210}$$

thus

$$\frac{\mu(R_t)}{\nu(R_t)} = 1 \qquad\qquad\qquad \lambda < c \tag{211}$$

$$\frac{\mu(R_t)}{\nu(R_t)} > 1 \qquad\qquad\qquad c < \sqrt{(1-q)}\lambda \tag{212}$$

$$\frac{\mu(R_t)}{\nu(R_t)} \leq 1 \qquad\qquad\qquad \sqrt{(1-q)}\lambda \leq c < \lambda \tag{213}$$

thus $c$ may always be chosen such that the (CSC) is less than $1$.

### F.3 PROOF OF THEOREM 5

With $\mu$ and $\nu$ given by equations (161) and (162) we have that

$$\lim_{c \to 0} \frac{\mu^2(R_t)}{\nu(R_t)} = \frac{2}{\pi} < 1 \tag{214}$$

Meanwhile, it can be seen that for all $R_t \geq 0$ $\mu_c^2(R_t)/\nu_c(R_t)$ is increasing and continuous in $c$. Since,

$$\lim_{c \to \infty} \frac{\mu_c^2(R_t)}{\nu(R_t)} = 1 \tag{215}$$

we are done.

### F.4 PROOF OF THEOREM 6

In light of Section F.2 above, we see that

$$\frac{\mu_c(R_t)^2}{\nu_c(R_t)} = 1 \qquad\qquad\qquad \lambda < c \tag{216}$$

$$\frac{\mu_c(R_t)^2}{\nu_c(R_t)} > 1 \qquad\qquad\qquad c < (1-q)\lambda \tag{217}$$

$$\frac{\mu_c(R_t)^2}{\nu_c(R_t)} \leq 1 \qquad\qquad\qquad (1-q)\lambda \leq c < \lambda \tag{218}$$

thus $c$ may always be chosen such that the (CCC) holds.

## G   THE RISK UNDER ANISOTROPIC DATA

In this section, we describe how to use equation (12) to solve for the risk. First, define

$$q_z(\mathbf{z}) = \mathbf{z}^T R(z; \mathbf{K})\mathbf{z}/2, \tag{219}$$

where $R(z; \mathbf{K}) = (\mathbf{K} - z\mathbf{I})^{-1}$ is the resolvent of $\mathbf{K}$. Using Itô's Lemma and the resolvent identity $R(z; \mathbf{K})(\mathbf{K} - z) = \mathbf{I}$.

$$
\begin{aligned}
\mathrm{d}q_z(\mathbf{\Theta}_t) = &- \eta(t) \left( \|\mathbf{\Theta}_t - \boldsymbol{\theta}^*\|^2 + 2zq_z(\mathbf{\Theta}_t) \right) \mu_{c(t)}(\mathcal{R}(\mathbf{\Theta}_t))\mathrm{d}t \\
&+ \frac{\eta^2(t)}{d} \operatorname{Tr}(\mathbf{K}R(z; \mathbf{K}))\nu_{c(t)}(\mathcal{R}(\mathbf{\Theta}_t))\mathrm{d}t + \mathrm{d}\mathcal{M}_t.
\end{aligned}
\tag{220}
$$

We shall let $Q_z(t)$ be the deterministic equivalent of this equation, that is

$$\frac{\mathrm{d}}{\mathrm{d}t}Q_z(t) = -\eta(t)\left(\mathcal{D}_t + 2zQ_z(t)\right)\mu_{c(t)}(R_t) + \frac{\eta^2(t)}{d} \operatorname{Tr}(\mathbf{K}R(z; \mathbf{K}))\nu_{c(t)}(R_t), \tag{221}$$

where (recalling $\Omega$ is the circle of radius 2)

$$D_t = \frac{-1}{2\pi i} \oint_\Omega Q_z(t)\mathrm{d}z \quad \text{and} \quad R_t = \frac{-1}{2\pi i} \oint_\Omega zQ_z(t)\mathrm{d}z.$$

These are analogues of the same formulas that hold exactly for $\mathcal{D}(\mathbf{\Theta}_t)$ and $\mathcal{R}(\mathbf{\Theta}_t)$ when replacing $Q_z$ by $q_z(\mathbf{\Theta}_t)$.

Now it is possible to precisely compare the solution of these ODEs to SGD, as the same machinery developed for Theorem 1 applies. In particular, Lemma 1 bounds the supremum difference $\sup_{z \in \Omega} |q_z(\mathbf{\Theta}_t) - Q_z(t)|$ (although now with $\mathcal{M}_t^{SDE} \equiv 0$). Hence, we conclude the following:

**Theorem 9.** *Suppose that Assumptions 1, 2 and 3 hold. Suppose that $\{\boldsymbol{\theta}_k\}$ is C-SGD. Let $\bar{c} = \sup_t c(t)$ and $\bar{\eta} = \sup_t \eta(t)$. There is a constant $\mathcal{C} = \mathcal{C}(\mathfrak{v}, (n/d), \bar{c}, \bar{\eta}, \|\boldsymbol{\theta}_0 - \boldsymbol{\theta}^*\|^2)$, a stochastic process $\mathcal{E}$, and a constant $m = m(\mathfrak{v})$ so that for any $1 \le u \le md$*

$$\sup_{0 \le k \le n} \left\| \begin{bmatrix} \mathcal{R}(\boldsymbol{\theta}_k) \\ \mathcal{D}(\boldsymbol{\theta}_k) \end{bmatrix} - \begin{bmatrix} R_{k/d} \\ D_{k/d} \end{bmatrix} \right\| \le \mathcal{C}\mathcal{E}(n/d)u \log(d)d^{-1/2}, \tag{222}$$

*with probability $1 - e^{-u}$ and provided the right hand side is less than 1. The stochastic process $\mathcal{E}$ is given by*

$$\mathcal{E}(t) = \exp\left( \int_0^t \frac{C\eta(s)^2\sigma \, \mathrm{d}s}{\sqrt{\mathcal{R}(\mathbf{\Theta}_s) + R_s}} \right)$$

*for an absolute constant $C > 0$. The constant $\mathcal{C}$ can be bounded by*

$$\mathcal{C} \le C\sqrt{n/d}\,\bar{\eta}\mathfrak{v}^2 \cdot ((1 + \|\boldsymbol{\theta}_0 - \boldsymbol{\theta}^*\|^2)\mathfrak{v}^2 + \bar{c}^2\sqrt{n/d}) \cdot \exp\left( C\max\{\bar{\eta}, \bar{\eta}^2\}(n/d) \right)$$

*for an absolute constant $C > 0$.*

We note that further details in this direction are shown in Collins-Woodfin et al. (2023).

### G.1   GETTING A SYSTEM OF ODES

We may use Equation (221) to get an equivalent coupled system of $d$ ODEs which can solve for $R_t$. First, we may diagonalize,

$$\mathbf{K} = \sum_{i=1}^{d} \lambda_i w_i w_i^T \qquad\qquad R(z; \mathbf{K}) = \sum_{i=1}^{d} \frac{1}{\lambda_i - z} w_i w_i^T \tag{223}$$

Where $\{\lambda_i\}_{i=1}^{d}$ and $\{w_i\}_{i=1}^{d}$ are the eigenvalues and eigenvectors of $\mathbf{K}$ respectively. Therefore,

$$Q_z(\mathbf{\Theta}_t) = \frac{1}{2} \sum_{i=1}^{\mathfrak{d}} \frac{1}{\lambda_i - z} \left\langle \mathbf{\Theta}_t - \boldsymbol{\theta}^*, w_i \right\rangle^2. \tag{224}$$

Define $v_i(t) = \left\langle \mathbf{\Theta}_t - \boldsymbol{\theta}^*, w_i \right\rangle^2 / 2$. Then, $R_t = \sum_{i=1}^{\mathfrak{d}} v_i(t)\lambda_i$ and $D_t = \sum_{i=1}^{\mathfrak{d}} 2v_i(t)$. Now, we can find a system of ODEs which describes the evolution of $\{v_i\}_{i=1}^{\mathfrak{d}}$.

Choose $\Omega_i$ to be a complex curve enclosing only the $i$-th eigenvalue of $\mathbf{K}$. Integrating over both sides of equation (221) and using Cauchy's integral formula, we see that

$$\frac{\mathrm{d}v_i}{\mathrm{d}t} = -2\eta(t)v_i\lambda_i\mu_{c(t)}(R_t) + \frac{\eta(t)^2}{d}\lambda_i\nu_{c(t)}(R_t)(R_t + \sigma^2/2), \quad \forall i \in [\mathfrak{d}]. \tag{225}$$

This final system of ODEs is used in all experiments to solve for $R_t$.

### G.2 Getting an Integral Equation

Using Equation (221) and an integrating factor we see that

$$\begin{aligned}
Q_z(t) =& Q_z(0)e^{-2z\Omega_t^c} + \frac{1}{d}\int_0^t \eta(s)^2\nu_{c(s)}(R_s)\,\mathrm{Tr}(\mathbf{K}R(z;\mathbf{K})e^{-2z(\Omega_t^c - \Omega_s^t)})(R_s + \sigma^2/2)\mathrm{d}s \\
& - \eta(t)D_t e^{-2z\Omega_t^c}\mu_{c(t)}(R_t)
\end{aligned} \tag{226}$$

where $\Omega_t^c = \int_0^t \eta(s)\mu_{c(s)}(R_s)\mathrm{d}s$ is the integrated clipped learning rate. Now, multiplying by $z$, integrating both sides around $\Omega$, and multiplying by $-1/2\pi i$, we get

$$R_t = \mathcal{R}(\mathbf{\Phi}_{\Gamma_T^c}^{\mathrm{gf}}) + \frac{1}{d}\int_0^t \tilde{\eta}^2(s)\nu_{c_s}\,\mathrm{Tr}(\mathbf{K}^2 e^{-2\mathbf{K}(\Gamma_t^c - \Gamma_s^c)})(R_s + \sigma^2/2)\mathrm{d}s, \tag{227}$$

where the first term is identified with gradient flow as in Paquette et al. (2022).

## H  Experimental details

### H.1  Clipped SGD and Homogenized Clipped SGD

The code to reproduce these results is available at https://github.com/nmarzz/clip.

#### H.1.1  Figure 1

The experiments creating Figure 1 were carried out on an M1 Macbook Air. Homogenized clipped SGD is solved via a standard Euler-Maruyama algorithm. The procedure for solving for the risk is described in Appendix G.

The synthetic data was generated with ambient dimension $\mathfrak{d} = 500$ while the intrinsic dimension is $d = 180$. For both experiments, $c_t = 0.9$, $\eta = 0.7$. We plot the $80\%$ confidence interval across 100 runs.

The CIFAR10 data was used to perform binary classification by regressing to $\pm 1$ labels. The data is split into classes 1 and 2 where class 1 contains: birds, cats, and dogs. Class 2 contains: trucks, ships, and planes. The data matrix $D$ is first passed through a random features model so that

$$D_{rf} = \tanh DA \tag{228}$$

where $A$ is a random features matrix of independent standard Gaussians. In order to estimate $\boldsymbol{\theta}^*$ the regression problem was first solved using Sci-kit learn (Pedregosa et al., 2011) and the resulting

solution was taken to be $\boldsymbol{\theta}^*$. The differences $\{y_i - \langle \boldsymbol{\theta}^*, \mathbf{x}_i \rangle\}$ for all $\mathbf{x}_i \in D_{rf}$ was then assumed to be the noise. A histogram of this noise is available in Figure 10a. The noise was then fitted to a Gaussian. Finally, $\eta = 0.1$ and $c = 0.5$ and the C-SGD plot represents the 80% confidence interval over 50 runs.

The Wikitext2 data was first processed for next-token-prediction. The data was tokenized and split into context lengths of at most 512. The data was then embedded using the Huggingface implementation of GPT-2 (Radford et al., 2019) and passed through the same random features model as with CIFAR10. We again used Sci-kit learn to estimate $\boldsymbol{\theta}^*$ and the noise terms whose histogram is available in Figure 10b. The noise resembled a mixture of 3-Gaussians and we therefore fit the noise to a Gaussian mixture model. In this case $\eta = 0.4$ and $c = 0.5$ and the C-SGD plot represents the 80% confidence interval over 50 runs.

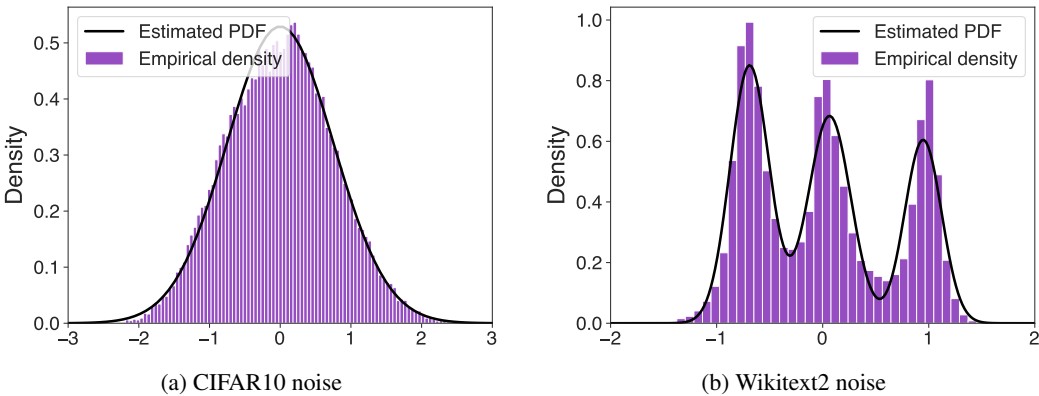

(a) CIFAR10 noise  (b) Wikitext2 noise

Figure 10: Histograms of the estimated noise distributions for the CIFAR10 and Wikitext2 datasets.

### H.1.2  FIGURE 4

The experiments creating were again carried out on an M1 Macbook Air. Numerical optimization of the max-(CCC) clipping schedule (Equation (18)) was done via the Nelder-Mead algorithm using standard python libraries.

### H.2  INTRINSIC DIMENSION EXPERIMENTS

The experiments in Appendix C.3 were carried out on 8 P100 GPUs trained in parallel with batch size 128. This allowed for efficient computation of the full batch Gauss-Newton operator norm via power iteration. Both networks were trained for 200 epochs. ResNet18 was trained with cosine learning rate decay (base learning rate 0.05), while ViT was trained with linear warmup for 2 epochs followed by a cosine learning rate decay (base learning rate 0.00625). Both networks used GELU activation function.

