# OpenReview forum: "To Clip or not to Clip: the Dynamics of SGD with Gradient Clipping in High-Dimensions"
_ICLR.cc/2025/Conference — ICLR 2025 Poster_

### Official Review · Reviewer_2Wzk · 2024-10-29

**Soundness:** 3
**Presentation:** 3
**Contribution:** 3
**Rating:** 6
**Confidence:** 2

**Summary:**

This paper provides a theoretical investigation into the effects of gradient clipping in high-dimensional settings. Specifically, the authors analyze how clipping impacts SGD in terms of stability, convergence, and optimization dynamics. They derive a deterministic equivalent for clipped SGD, named Clipped Homogenized SGD, which approximates the risk evolution via a system of Ordinary Differential Equations (ODEs) under high-dimensional assumptions. Key findings indicate that while clipping does not improve performance with Gaussian noise, it can enhance stability and convergence under other noisy conditions. The paper also proposes a heuristic clipping schedule for near-optimal performance with minimal hyperparameter tuning.

**Strengths:**

1. The study provides a new theoretical perspective on gradient clipping, a widely used technique in training deep learning models, by framing it in high-dimensional analysis. This theoretical foundation is novel and helps bridge a gap in understanding the dynamics of SGD under clipping.

2. The paper is well-organized, with a clear presentation of problem setup, definitions, and theoretical results. The analysis is rigorous, covering a range of noise distributions and demonstrating the proposed model’s effectiveness across synthetic and real-world datasets

**Weaknesses:**

1. The analysis is specific to linear regression, which may limit the generalizability of the findings to other types of models. Insights into extending these results or experiments with more complex models, such as neural networks, could enhance the broader relevance of the work.

2. The descent and variance reduction factors play a central role in understanding the stability and effectiveness of clipping. However, the paper lacks a practical interpretation or intuition for these factors, which might make it challenging for practitioners to apply these insights beyond the linear regression context. Further discussion on the role of these factors in practical scenarios, perhaps with illustrative examples, would enhance accessibility and the broader applicability of the results.

**Questions:**

1. Can the authors provide more insights into how sensitive the results are to the hyperparameter choice in the proposed clipping heuristic? Would small variations in the threshold significantly affect the stability or convergence?

2. Could the authors discuss the potential for extending their theoretical findings beyond linear regression to other models, such as generalized linear models? Are there specific challenges or limitations in applying their stability analysis to non-linear models or more complex loss functions?

---

> ### Author Response · Authors · 2024-11-15
>
> Thank you very much for the constructive feedback. Below we would like to address your concerns and elaborate on some of our key points.
>
> ## Weaknesses
>
> 1. Indeed it is important to understand how clipping will affect training in other settings perhaps over other convex losses. We believe that this is possible in a manner similar to [1] which provides results for unclipped SGD in terms of the ambient dimension (rather than the intrinsic dimension) over a class of non-quadratic losses. However, the analysis required for these settings is rather long and so we felt it important that a full understanding of the quadratic setting was important.
>
> Additionally, understanding the behavior over quadratic losses can tell a lot about even non-quadratic cases (see for example [2]). Even with a description for the behavior of clipping for non-quadratic problems the most natural question to ask would be if it differs significantly from the quadratics.
>
> 2. We have attempted to give an intuition for these factors after introducing them (see lines 155-161). We also give upper and lower bounds for these factors such that up to constants our bounds are equivalent (Equation 10a, 10b). We’ll elaborate on the intuition now and improve the clarity in revision:
> - The mean reduction factor measures how much, in expectation, clipping reduces the length of the gradient.
> - The variance reduction factor measures how much, in expectation, clipping reduces the variance of the norm-squared of the gradient.
>
> In our original submission, we showed that these factors can indeed be estimated in an online manner in practical neural networks, using statistics of the stochastic gradients already computed during training (Appendix C.2). We demonstrated their computation on ResNet18 and ViT. We think detailed exploration of $\mu$, $\nu$, and their usefulness for setting practical clipping schedules is an exciting avenue for future work.
>
>
> ## Questions:
>
> 1. We think that this sensitivity question is interesting and have made some progress to explore it. Given your suggestion, we’ve provided some results in Appendix C.4 showing the relative insensitivity of the heuristic to changes in the optimal hyper-parameter.
>
> 2. We hope we have address the question of extending our result in the text above responding to listed weaknesses.
>
>
>
> **Reference:**
> [1] Collins-Woodfin et al., "Hitting the high-dimensional notes: An ode for SGD learning dynamics on GLMs and multi-index models," 2023.
> [2] Hui and Mikhail Belkin. Evaluation of neural architectures trained with square loss vs cross entropy in
> classification tasks. ICLR2021.

---

> > ### Comment · Reviewer_2Wzk · 2024-11-25
> >
> > Thank you for your detailed response. I have no other concerns and will keep my rating

---

### Official Review · Reviewer_g87A · 2024-10-30

**Soundness:** 3
**Presentation:** 2
**Contribution:** 3
**Rating:** 6
**Confidence:** 3

**Summary:**

The paper studies the widely used gradient clipping technique in SGD under quadratic settings. Under specific assumptions, the paper develops a theoretical analysis of the training dynamics by proposing an SDE C-HSGD and proving its connection to clipped SGD. Based on this, the paper shows that when clipping can help convergence and stability, providing insights into understanding the clipping technique.

**Strengths:**

1. To my knowledge, this is a novel perspective looking into clipping, which is interesting and aligns with the common belief that clipping helps when noise is heavy-tailed.

2. The theory seems to be clear and complete.

**Weaknesses:**

1. Although the theory looks promising, I think it's still restrictive due to the settings it applies:
    - As the author also mentioned in the paper, one of the key benefits of clipping is that it helps SGD converge in generalized smooth settings, which cannot be reflected by quadratics.
    - The theory typically requires the data $x$ to be Gaussian, or close to Gaussian as in Appendix A. This is still a restrictive assumption.

2. I kindly suggest the authors gather more notation definitions in Section 2, as there are still new notations in the following sections, which may make it inconvenient to follow and refer to when reading.

**Questions:**

1. More intuitive explanations for the connection between C-SGD and S-HSGD are welcomed.

2. As also noted in the weakness part, I think possibly the authors can discuss how the analysis in the Gaussian quadratic settings may have impacts on broader machine learning applications.

3. Do you also take stability into consideration when comparing C-SGD and SGD in Section 5?

---

> ### Author Response · Authors · 2024-11-15
>
> Thank you for the constructive feedback. Below we would like to address your concerns and elaborate on some of our key points.
>
> ## Weaknesses
>
> 1. Indeed it is important to understand how clipping will affect training in other settings perhaps over other convex losses. We believe that this is possible in a manner similar to [1] which provides results for unclipped SGD in terms of the ambient dimension (rather than the intrinsic dimension) over a class of non-quadratic losses. However, the analysis required for these settings is rather long and so we felt it important that a full understanding of the quadratic setting was important.
>
> Additionally, understanding the behavior over quadratic losses can tell a lot about even non-quadratic cases (see for example [2]). Even with a description for the behavior of clipping for non-quadratic problems the most natural question to ask would be if it differs significantly from the quadratics.
>
> 2. Certainly it is desirable to precisely understand the form of the $\mu$ reduction factor for non-Gaussian data. However, it is likely that it is not meaningfully very different from the Gaussian case. This might be seen in two ways:
> - We provide evidence that the theory is effective in reproducing real-world results across two distinct settings: vision and NLP with the CIFAR10 and Wikitext2 datasets.
>
> - It seems that a universality principle is likely to apply. Universality is a common phenomenon seen in high-dimensionals where large-scale behaviors do not depend on the precise distributions of the individual components. As a simple example, the gradient coefficient $\ell_{\boldsymbol{\theta}}$ should see CLT effects as (ambient) dimension grows.
>
> 3. Our results are proven to hold for subgaussian data (Theorem 7). Subgaussians need not be “close” to Gaussians save for their tail behavior. For example, _any_ distribution with finite support is subgaussian. This includes any vision dataset (consider RGB images whose values range from 0 to 255) or any tokenized text dataset with a finite vocabulary that does not scale with dimension.
>
> 4. There is a lot of notation to handle in this paper. We’ll work to add a section in the Appendix to help ease the struggle.
>
> ## Questions:
> 1. Roughly the coefficients of C-HSGD are chosen to match the first and second moments of the risk (or distance to optimality) under C-SGD. Importantly though they are not exactly the same. The high-intrinsic-dimension limit allows for a significant simplification of these terms as any errors introduced by the simplifications vanish in the limit. That said, it is correct, to a first approximation, to think of matching first and second moments.
>
> 2. We believe that rigorously finding the optimal clipping schedule in a simpler setting is the first step to developing better heuristics in the practical setting. For example, neural networks are known to linearize as their width approaches infinity [3] implying that we could describe the behaviour of a wide neural network trained with C-SGD.
>
> We found that the reduction factors $\mu$ and $\nu$ are sufficient to compute optimal clipping schedules in our setting. We hypothesize that $\mu$ and $\nu$ play an important role in the more general non-linear setting as well, and in our original submission we described an efficient way to estimate them online for practical networks. We also found simpler heuristics for the optimal clipping schedule based on the risk and noise estimation. We hope to investigate these quantities in practical models in followup work.
>
>
> 3. Stability is considered but only implicitly. If the CCC is satisfied the CSC must be satisfied as well (since $0 \leq \mu \leq 1$). Therefore in all cases in Section 5 where the CCC is met the CSC is implicitly satisfied as well.
>
>
>
> **References:**
>
> [1] Collins-Woodfin et al., "Hitting the high-dimensional notes: An ode for SGD learning dynamics on GLMs and multi-index models," 2023.
>
> [2] Hui and Mikhail Belkin. Evaluation of neural architectures trained with square loss vs cross entropy in
> classification tasks. ICLR2021.
>
> [3] Jacot et al., “Neural Tangent Kernel: Convergence and Generalization in Neural Networks“ NeurIPS 2018.

---

> > ### Comment · Reviewer_g87A · 2024-11-22
> >
> > Thanks for the reply. I will keep the score as is.

---

### Official Review · Reviewer_dT4j · 2024-11-03

**Soundness:** 3
**Presentation:** 3
**Contribution:** 2
**Rating:** 6
**Confidence:** 4

**Summary:**

The paper studies the properties of clipped SGD for linear regression with mean squared error loss under specific data and noise distributions. The authors compare the parameters learned from clipped SGD (C-SGD) to the solution of a stochastic differential equation (C-HSGD), demonstrating their close alignment in terms of risk and optimality gap. They also derive an ODE to describe risk evolution. Additionally, learning stability (CSC) and instantaneous risk descent (CCC) are analyzed through descent and variance reduction factors. The authors show that under Gaussian noise, clipped SGD is inferior to unclipped SGD, but under other noise distributions, it improves both performance and stability. Theoretical results are supported by experiments.

**Strengths:**

The paper is very well-written, and the problem it addresses is important for large-scale learning settings.  Particularly, gradient clipping is an important  technique that proves useful in deep learning, which makes it crucial to understand its effects. This paper provides a rigorous characterization of clipped SGD in linear regression, focusing on the descent and variance reduction factors. The authors also propose a learning rate schedule that ensures clipped SGD performs at least as well as standard SGD, offering practical insights for optimization.

**Weaknesses:**

* The problem setting is restricted to linear regression with mean squared loss  and the proposed heuristic seems to be meaningful only when the features follow a Gaussian distribution.

* Calculating the descent and variance reduction factors in more general settings appears challenging, since it involves numerical optimization at every step, which limits the practical applicability of the proposed learning rate scheduling and criteria.

**Questions:**

* In the conclusion, the authors suggest extending the analysis to more complex losses. What are the bottlenecks for such an analysis?
* The heuristic presented in the paper appears to be limited to normally distributed data due to the validity of Eqs. (10a) and (10b). Is it possible to extend this approach to a more general setting, such as sub-Gaussian distributions?
* While the authors used real-world data to validate the C-HSGD approximation, they did not evaluate the heuristics on real-world data (though the setup for Figure 4 is somewhat unclear, please correct me if I am mistaken). Could the authors clarify why this aspect was not explored?

Minor:
* In Theorem 1, the subscript for $\theta_{sd}$  in the denominator of the definition of $\mathcal{E}$ should include a ceiling (or floor).

---

> ### Author Response · Authors · 2024-11-15
>
> Thank you for your feedback, and for your careful reading of our paper. You’re correct about the missing floor and we are happy to address your major concerns and clarify some points.
>
> We see three main points made in your weaknesses and questions that we would like to address separately:
>
>
> **Restriction to linear regression with MSE loss**
>
> Though our initial work was restricted to MSE loss, we believe it’s possible to extend our work to other losses (including cross-entropy loss), and some classes of non-linear models. Recent work established a high-dimensional theory for streaming SGD with Gaussian data on a large class of loss functions, and a broader class of models [1]. This approach required large ambient dimension, and the assumption that intrinsic dimension $\approx$ ambient dimension.
>
> We believe that a similar approach could be used to understand clipped SGD as well. However this requires very detailed technical work ([1] is 98 pages!) that benefits from an understanding of simpler cases. We hope to pursue this avenue in the near future.
>
> We also believe analyzing $\mu$ and $\nu$ in more realistic models is tractable and promising (see Appendix C.2 of our original submission); this is another area of future work.
>
>
> **Applicability of the heuristics to real data**
>
> We thank the reviewer for bringing up this point. We have revised our submission to test the heuristics on real-world datasets (Appendix C.4, on wikitext2). We find that the heuristics work well even for non-Gaussian data. Indeed we find that if the mean reduction factor $\mu$ is well-characterized by the probability of clipping, the heuristic proof (Theorem 8 and Corollary 1) extends to any data distribution that is sufficiently regular around 0 (formally, any distribution with an integrable characteristic function and positive density at 0).
>
> It may be possible to derive a universality result which extends the heuristic to a large class of subgaussian data distributions; the clipping functions behavior is determined by the relationship between the gradient norm and clipping threshold. In high dimensions the gradient norm becomes more and more independent of distributional details, increasing the utility of our heuristic. We leave exploration of this universal behavior for future work.
>
>
>
> **Calculating the mean reduction factors $ \mu $ and $ \nu $ in general settings**
>
> In our original submission, we described an efficient online estimator of $\mu$ and $\nu$ which uses gradient statistics already collected during training (Appendix C.2). We implemented and ran the estimator during training runs of ResNet18 and ViT, to prove that the estimator works.
>
> We neglected to mention that the estimators can be run in parallel for many values of the clipping threshold c, with negligible overhead. This lets us cheaply optimize the CCC (and therefore the clipping threshold) at every step; we simply maximize using the estimators over our fixed grid of c.
>
> For MSE loss we can also use our heuristics from Equation 10 as estimators. We believe other estimators can be derived for common losses like cross-entropy.
>
>
>
>
> **Reference:**
> [1] Collins-Woodfin et al., "Hitting the high-dimensional notes: An ode for SGD learning dynamics on GLMs and multi-index models," 2023.

---

> ### Comment · Reviewer_dT4j · 2024-11-22
>
> Thank you to the authors for their detailed responses. I agree that the universality of the heuristics seems reasonable under mild conditions. Based on the clarifications during the rebuttal, I’m raising my score from 5 to 6. Good luck!

---

### Official Review · Reviewer_cyxq · 2024-11-05

**Soundness:** 3
**Presentation:** 3
**Contribution:** 3
**Rating:** 8
**Confidence:** 3

**Summary:**

This paper analyzes SGD with gradient clipping by constructing an analogous Stochastic Differential Equation (SDE) and studying its solution. The SDE developed by the authors follows the development in [1] with the addition of two factors $\mu_c$ and $\nu_c$ that capture the effect of gradient clipping. The authors show that their SDE is close to the SGD trajectory and also conduct a stability analysis to obtain a criterion for when clipped SGD is more or less stable than its unclipped version (and allows for larger learning rates). The authors study different noise conditions (Gaussian and Rademacher-type) noise and show that the clipping threshold can be set to obtain larger learning rates. Finally this paper explores the conditions under which clipped SGD can achieve lower risk values than unclipped SGD.

[1] Paquette, C., Paquette, E., Adlam, B., & Pennington, J. (2022). Homogenization of SGD in high-dimensions: Exact dynamics and generalization properties. arXiv preprint arXiv:2205.07069.

**Strengths:**

The theoretical analysis seems strong and the authors provide extensive technical proofs for their theoretical results. While I did not go through all the proofs to verify their correctness, the analysis seems correct. The authors go beyond just showing a correspondence between clipped SGD and the C-HSGD stochastic differential equation to provide some applications of their result to stability analysis and to study when clipping helps in SGD.

**Weaknesses:**

1. The theoretical analysis seems to mostly be an application of previous results obtained in [1]. The new machinery that is used here is Stein's lemma to obtain bounds on the quantities $\mu_c$ and $\nu_c$ that characterize the effects of clipping SGD. This only allows the authors to characterize $\mu_c, \nu_c$ for gaussian data, even though they are able to show the correspondence of the SDE to clipped SGD with non-gaussian data. The technical contribution is arguably only incremental.

2. The condition for testing whether clipped SGD reaches lower risk values (CCC) seems to be hard to evaluate. It involves estimating whether $\mu_c^2 / \nu_c > 1$ anywhere along the trajectory of the SDE. This seems to require solving the SDE to estimate whether clipped SGD is beneficial, as well as estimating the threshold $c$ at which to clip the gradients. This may be harder than simply solving the original least squares optimization problem.

3. While the authors explore how their results extend to the setting of training deep networks, the arguments are made in the regime where deep networks can be approximated well by linear models (the NTK regime). NTK models do not perform as well as deep networks that learn features and that cannot be approximated as linear models.

[1] Paquette, C., Paquette, E., Adlam, B., & Pennington, J. (2022). Homogenization of SGD in high-dimensions: Exact dynamics and generalization properties. arXiv preprint arXiv:2205.07069.

**Questions:**

1. The role of CSC is still unclear to me. Can the authors provide a demonstration of how clipped SGD allows for larger learning rates. None of the figures seem to illustrate how clipped SGD converges faster than unclipped SGD. In case I have missed this please direct me to where I can find this.

2. While there is a figure demonstrating the correspondence of clipped SGD and the SDE with heavy tailed Cauchy noise, there are no experiments comparing unclipped and clipped SGD in that condition. Does clipped SGD converge faster in that situation? Can this be demonstrated in an experiment?

---

> ### Author Response · Authors · 2024-11-15
>
> Thank you for your thoughtful review. We’re glad you appreciate our work and the value of our results to understanding and improving the usage of gradient clipping. Below, we address your specific comments and concerns in detail.
>
>
> ## Weaknesses
>
>
> 1. Our approach does leverage foundational techniques from Paquette et al. (2022) and builds on their homogenization framework. However, we provide a novel technical contribution by considering learning rate scalings by the intrinsic dimension, better reflecting the truth and providing a non-asymptotic statement. This is a non-trivial contribution even for unclipped SGD. When the data lie in a low dimensional subspace SGD does not concentrate. In comparison to previous work our paper captures this behavior.
>
> Additionally, our theory does apply beyond the Gaussian setting (see Theorem 7) and it is only the identification of $\mu$ with the probability of not clipping that requires Gaussianity.
>
>    Together, these contributions provide a more complete picture of how clipping affects SGD, particularly in high-dimensional, noisy environments, than what was previously known.
>
>
>
>
> 2.  We agree that estimating whether CCC $> 1$ can be challenging *a priori*. The aim of this result (Theorem 4) is to provide a theoretical tool, along with intuitive guidance for when clipped SGD is effective. Interpreting this CCC result leads to the guiding principle: if the variance of the gradients can be drastically reduced relative to the probability of clipping at any time during training (for any risk value one is likely to achieve) then clipping will be effective. Our result formalizes this idea.
>
> Additionally, this criteria is not so difficult to measure in practice. The definitions of $\mu$ and $\nu$ in Equation (7) requires an estimation of stochastic gradients and their norms; both of these are *already* computed during training. Therefore a sort of online estimation of $\mu$ and $\nu$ are readily available during training at negligible cost. We have elaborated on this point and performed preliminary experiments in Appendix C.2. A comprehensive study implementing these results is reserved for future work.
>
>
> 3.  Your observation about the limitations of applying our results to the NTK (Neural Tangent Kernel) regime is well-taken. We chose this linear approximation framework for tractability and to lay a foundation for understanding how clipping interacts with learning dynamics in a simplified setting. However, it is clear that linear models cannot fully capture the entire dynamics of SGD.
>
> We believe it is important to first check that this simpler setting provides meaningful results. Having laid the groundwork, future results regarding the effects of non-linearity can be compared in relation to this null model of linear models.
>
> ## Questions
>
> 1.  It should be noted that if the CCC > 1 then the CSC > 1 as well (since $0 \leq \mu  \leq 1$). For this reason we chose to aim our discussion at the former. If the CSC > 1 but the CCC is not (at every point along the training path), then clipped SGD won’t converge faster than unclipped SGD. In this case, clipped SGD would have the (still noteable) effect of increasing robustness to a sub-optimal choice of learning rate, increasing the rate of values under which the optimizer will converge.
>
> 2. We did not include explicit comparisons between unclipped and clipped SGD under heavy-tailed noise, such as Cauchy noise, because unclipped SGD diverges. This can be seen given the explicit dependence of the risk on the variance of the noise term. A proof of this fact has been given by [1] in Remark 1.
>
>
> Thank you again for your feedback and for highlighting areas where we can improve our submission.
>
>
>
> [1] Zhang, J., Karimireddy, S., Veit, A., Kim, S., Reddi, S., Kumar, S., & Sra, S. (2020). Why are Adaptive Methods Good for Attention Models?. In Advances in Neural Information Processing Systems (pp. 15383–15393)

---

> > ### Comment · Reviewer_cyxq · 2024-11-26
> >
> > Thanks for your detailed response. I don't think I need to update my rating since my questions were mostly minor points.

---

### Author Response · Authors · 2024-11-15
**Global Response to Reviewers**

We would like to thank all Reviewers for their constructive feedback. We appreciate the opportunity to address concerns raised and clarify our contributions. Below, we summarize and address the main recurring concerns across the reviews:


**Generality of results beyond linear regression**

   Several reviewers noted that our analysis is limited to linear regression with mean squared error loss, raising concerns about the broader applicability of our findings.

We intentionally focused on this simple setting to provide a solid and rigorous foundation for building intuition about gradient clipping. Understanding optimization under a quadratic loss serves as a baseline to sanity-check more complicated losses.

There is a clear path to analysis on non-quadratic losses; for ordinary SGD, [1] provides a way to extend high-dimensional analysis to a broad family of loss functions including cross-entropy loss; analogous methods should work for clipped SGD. This approach would also allow analysis of some non-linear models, including certain classes of neural networks. The work in [1] is extremely technical; therefore, we chose to establish the theory in the simpler case first, in order to better inform the generalization of the methodology to more complicated losses.

We also believe that our insights are relevant for more realistic non-linear models. Understanding the connections will require a detailed empirical study. We are encouraged by the fact that our key quantities — the mean and variance reduction factors $\mu$ and $\nu$ — have tractable estimators in the full non-linear setting (Appendix C.2). We leave this study for future work.

**Assumptions on the data**

Some reviewers raised concerns about our data modeling assumptions. The majority of our analysis holds over subgaussian data — a large class which contains many distributions of interest, such as those with finite support, including tokenized NLP data with finite vocabulary.

In our original submission we provided experimental evidence that our results are informative for real data in both vision and NLP contexts (Figure 1) — as expected since those data are subgaussian. Reviewers expressed concern that our heuristics for optimal clipping depended on the Gaussian formulation of the mean reduction factor $\mu$. We have added additional experiments (Appendix C.4) which show that our optimal clipping heuristic is also effective for real, non-Gaussian data.



**Practical utility of descent and variance reduction factors**

There was also concern about the interpretability and practical utility of the descent ($\mu$) and variance reduction ($\nu$) factors. The interpretation is found in lines 155-161 of our paper and can be summarized as follows:
- The mean reduction factor ($\mu$) measures how much, in expectation, clipping reduces the length of the gradient.
- The variance reduction factor measures how much, in expectation, clipping reduces the variance of the norm-squared of the gradient.
In our setting we also derived simple upper and lower bounds of these factors, with respect to the loss and noise statistics which are easy to calculate and interpret (Equations 10a, 10b). This gave a heuristic for the optimal clipping schedule (Equation 25). Following reviewer comments we have added evidence that the heuristic is relatively _insensitive_ to the constants in the bounds (Appendix C.4).

In our original submission we also included a methodology to compute the reduction factors in non-linear settings using an online estimator (Appendix C.2, ResNet18 and ViT). We neglected to mention that this estimator can track $\mu_{c}$ and $\nu_{c}$ for many $c$ values in parallel at virtually no cost, which improves its utility for choosing a good instantaneous clipping threshold.




_(Revisions in the paper are highlighted in red so they may be seen easily)_


[1] Collins-Woodfin et al., "Hitting the high-dimensional notes: An ode for SGD learning dynamics on GLMs and multi-index models," 2023.

---

### Author Response · Authors · 2024-11-15

We noticed a minor bug in our code for the heuristic optimal clipping schedule. As a result we have updated Figure 4. This has changed the details of the figure but the overall message remains unaffected.

---

### Author Response · Authors · 2024-11-22
**Any further questions?**

As the rebuttal period is nearing its conclusion, please let us know if there are any other points we can clarify!

---

### Author Response · Authors · 2024-11-26
**Thank you to the reviewers**

We'd like to thank the reviewers for taking the time to review our work.

We appreciate the reviewers' positive feedback and recognition of the strengths and contributions of our work. Comments largely highlighted minor points, and we are pleased that no substantial changes were necessary.

Our manuscript has been updated with one experiment verifying the effectiveness of our heuristic clipping schedule on real world data, further validating our findings. Thank you for your constructive feedback and for acknowledging the clarity and utility of our analysis of clipped SGD.

---

### Meta-Review · Area_Chair_Um4U · 2024-12-13

**Metareview:**

The paper studies the properties of clipped SGD for linear regression with mean squared error loss under specific data and noise distributions. The authors have addressed all the reviewers' concerns. All the reviewers are happy to recommend the acceptance of the paper. Please incorporate all the comments and discussions into the final version.

**Additional Comments On Reviewer Discussion:**

No concerns.

---

### Decision · Program_Chairs · 2025-01-22

Accept (Poster)